



# Comparing the transport-limited and $\xi - q$ models for sediment transport

Jean Braun[1,2]

[1]Helmholtz Centre Potsdam, GFZ German Research Centre for Geosciences, Potsdam, Germany
[2]Institute of Earth and Environmental Sciences, University of Potsdam, Potsdam, Germany

**Correspondence:** Jean Braun (jbraun@gfz-potsdam.de)

**Abstract.** We present a comparison between two of the most widely used reduced-complexity models for the representation of sediment transport and deposition processes, namely the transport limited (or $TL$) model and the under-capacity (or $\xi - q$) model more recently developed by Davy and Lague (2009). Using both models, we investigate the behavior of a sedimentary continental system fed by a fixed sedimentary flux from a nearby active orogen though which sediments transit to a fixed base

level representing a large river, a lake or an ocean. Our comparison shows that the two models share the same steady-state solution, for which we derive a simple 1D analytical solution that reproduces the major features of such sedimentary systems: a steep fan that connects to a shallower alluvial plain. The resulting fan geometry obeys basic observational constraints on fan size and slope with respect to the upstream drainage area, $A_0$. The solution is strongly dependent on the size of the system, $L$, in comparison to a distance $L_0$ that is determined by the size of $A_0$ and gives rise to two fundamentally different types of

sedimentary systems: constrained system where $L < L_0$ and open systems where $L > L_0$. We derive simple expressions that show the dependence of the system response time on the system characteristics, such as its length, the size of the upstream catchment area, the amplitude of the incoming sedimentary flux and the respective rate parameters (diffusivity or erodibility) for each of the two models. We show that the $\xi - q$ model predicts longer response times, which we relate to its greater efficiency at propagating signals through its entire length. We demonstrate that, although the manner in which signals propagates through

the sedimentary system differs greatly between the two models, they both predict that perturbations that last longer than the response time of the system can be recorded in the stratigraphy of the sedimentary system and in particular of the fan. Interestingly, the $\xi - q$ model predicts that all perturbations in incoming sedimentary flux will be transmitted through the system whereas the $TL$ model predicts that rapid perturbations cannot. We finally discuss why and under which conditions these differences are important and propose observational ways to determine which of the two models is most appropriate to

represent natural systems.

## 1   Introduction

Sedimentary basins contain the record of Earth's past tectonic and climatic histories. To untangle this record, scientists often rely on the use of numerical models that simulate the physical processes controlling sediment production, transport and deposition. Models are commonly used to characterize the response of sedimentary systems to external forcing in the source area





(change in tectonic uplift rate or in rainfall intensity) or in the depositional environment (variations in sea level). In particular whether perturbations can propagate across so-called "source-to-sink" systems remains an open question (Romans et al., 2016; Tofelde et al., 2021) that models have attempted to answer (Castelltort and Van Den Driessche, 2003; Simpson and Castelltort, 2006; Armitage et al., 2011, 2013; Mouchené et al., 2017).

Traditionally, sediment transport has been represented using a non-linear diffusion equation assuming that the process is
limited by the transport capacity of rivers (the main transport agents) that is assumed to be proportional to slope and discharge and to other factors, including grain size (Henderson, 1966). We will name this model the transport-limited or $TL$ model. Recently Davy and Lague (2009) introduced a new model (which they named the $\xi - q$ model) to represent the competition between sediment production (erosion), transport and deposition in fluvial systems. Improving on the work from previous authors (Kooi and Beaumont, 1994), their main purpose was to produce a model that could account for the transition from
detachment-limited to transport-limited behaviors of mountain channels. In recent years, the model has also been used to study sedimentary systems away from the orogenic area (Carretier et al., 2016; Shobe et al., 2017; Yuan et al., 2019) and this has led to attempts (Guerit et al., 2019) to quantify the value of the main model parameter, $\xi$, originally described as a characteristic distance for deposition that depends on discharge but later remapped into the inverse of a rate (Carretier et al., 2016) or a dimensionless number (the $\Theta$ parameter of Davy and Lague (2009) or the $G$ parameter of Yuan et al. (2019)).

Although Davy and Lague (2009) described in great detail the behavior of their model, including the conditions that favor transport-limited over detachment-limited behavior or the response time of a system obeying their formulation to both long and short-term variations in uplift rate, the behavior of the model in a purely depositional environment has not been studied thoroughly. We believe it is, however, essential that such an analysis be made in order to validate this model or, at minimum, to understand its limits of applicability and, ultimately, adequately interpret the predictions that might be made by using it in
future work. This is what we propose to do here as well as comparing its predictions to the traditional non-linear diffusion approach or $TL$ model

It is important, however, to keep in mind that the $\xi - q$ model behavior asymptotically tends towards that of the $TL$ model for small values of the depositional length $\xi$ or, more correctly, for large values of the $\Theta$ dimensionless number introduced by Davy and Lague (2009) or the $G$-factor introduced by Yuan et al. (2019). Even though one of them, the $\xi - q$ model 'contains'
the other, we will compare the two models as independent of one another rather than comparing the effect of an infinite value of the $Theta$ dimensionless number, mostly for practical reasons (as we do not know how large a value of $\Theta$ to use for the $\xi - q$ model to behave exactly like the $TL$ model) but also because the $TL$ model existed before its generalization was introduced.

Finally, to keep this study as general as possible, we will target the behavior of both models at a range of scales from that of the fan area next to the orogenic system to that of the neighboring large alluvial plain.



## 2 Method

### 2.1 The two models

Traditionally, the transport of sediment by rivers has been modeled using the Transport Limited (or $TL$) model (Henderson, 1966). In the $TL$ model a river is assumed to transport sediment at its transport capacity. The transport capacity or optimum flux of sediment, $q$ (expressed in m$^2$ yr$^{-1}$), is assumed to be proportional to local topographic slope, $S$ (expressed in m m$^{-1}$), and specific discharge, $q_w$ (expressed in m$^2$ yr$^{-1}$), raised to some powers, $m+1$ and $n$:

$$q \propto q_w^{m+1} |S|^n \tag{1}$$

Specific discharge will be assumed to be the product of upstream drainage area, $A$ (expressed in m$^2$), by net precipitation rate $\nu$ (dimensionless) relative to some reference value that is commonly inserted into a rate parameter or transport coefficient, $K_d$ (expressed in m$^{1-m}$yr$^{-1}$), divided by the flood-plain width, $w$ (expressed in m) to yield:

$$q = K_d \left(\frac{A\nu}{w}\right)^{m+1} |S|^n = \frac{K_d}{w^{m+1}} (A\nu)^{m+1} |S|^n \tag{2}$$

Conservation of mass leads to the following evolution equation for surface elevation, $h$ (expressed in m):

$$\frac{\partial h}{\partial t} = \frac{\partial}{\partial x} \frac{K_d}{w^{m+1}} (A\nu)^{m+1} \left|\frac{\partial h}{\partial x}\right|^n \tag{3}$$

where $x$ is the direction of flow in the river (expressed in m), $t$ is time (expressed in yr) and noting that $S = \frac{\partial h}{\partial x}$. Note that we have assumed that there is only one material that is transported, deposited and potentially eroded, such that we do not need to worry about density differences between what is transported and eroded/deposited off the river bed. We will call Equation 3 the $TL$ equation.

The $\xi - q$ model (Davy and Lague, 2009) assumes that the rate of change of topographic height is the sum of two terms, one representing erosion and the other deposition. Erosion rate, $\dot{e}$, is assumed to be governed by the stream power law (SPL) and thus proportional to the product of specific discharge and slope raised to some power (Howard and Kirby, 1983; Whipple and Tucker, 1999):

$$\dot{e} \propto q_w^m |S|^n \tag{4}$$

while deposition rate, $\dot{d}$, is assumed proportional to the ratio of upstream-integrated sedimentary flux and a deposition length that depends on specific discharge, $\xi(q_w)$ (Davy and Lague, 2009):

$$\dot{d} \propto \frac{q}{\xi(q_w)} \tag{5}$$

We will follow Davy and Lague (2009) and assume that $\xi$ is given by

$$\xi(q_w) = \frac{q_w}{d^* v_s} \tag{6}$$





where $v_s$ is the net settling velocity of sediment particles (i.e., taking into account turbulence) and $d^*$ a dimensionless parameter characterizing the distribution of particles in the river (it is the ratio of the water column height by the thickness of the actively transporting layer). This leads to the following evolution equation:

$$\frac{\partial h}{\partial t} = -K_f (\frac{A\nu}{w})^m |S|^n + \frac{Gw}{A\nu} q = -\frac{K_f}{w^m}(A\nu)^m |S|^n + \frac{Gw}{A\nu}(q_0 - \int_0^x \frac{\partial h}{\partial t}\, dx) \tag{7}$$

where $K_f$ is the erodibility coefficient that has units of $\mathrm{m}^{1-m}\ \mathrm{yr}^{-1}$and $G$ is a dimensionless parameter defined as:

$$G = \frac{d^* v_s}{\nu_0} \tag{8}$$

where $\nu_0$ is mean precipitation rate. The parameter $G$ was proposed by Yuan et al. (2019) and is equivalent to the parameter $\Theta$ introduced by Davy and Lague (2009). In their implementation of the $\xi - q$ model, Carretier et al. (2016) used a parameter

relating the depositional length to specific discharge that they call $\zeta$ and has the dimensions of the inverse of a velocity ([T]/[L]). It is related to the dimensionless parameter, $G$, used here by the following relationship:

$$G = \frac{1}{\zeta \nu_0} \tag{9}$$

Davy and Lague (2009) estimate that $\Theta$ (or $G$) is likely to be greater or equal to one, depending on grain size, rainfall intensity and variability (Guerit et al., 2019). These authors use the change in channel slope at the orogenic front (where uplift rate

vanishes abruptly) to estimate the value of $G$. Compiling observations from many sedimentary systems, they estimate that $G$ must be in the range [1-2].

Note that in both equations 3 and 7, we will assumed that the floodplain width, $w$, is constant. As shown by Nardi et al. (2006), flood plain width varies as a weak function of drainage area, i.e., $w \propto A^\theta$, with $\theta \approx 0.2 - 0.3$. However, one could consider $w$ to be an averaged value of the floodplain width for the system we consider and that its variation with drainage area

or discharge is factored in the value of the exponent $m$, as commonly done/assumed.

## 2.2 Experimental setup

To compare the behavior of these two equations, we will use a very simple setup (Figure 1) in which an initially flat ($h = 0$) surface of length $L$ accumulates sediment brought at a constant flux, $q_0$, across its left-hand side boundary at $x = 0$. The drainage area will be assumed to obey Hack's law:

$$A(x) = A_0 + kx^p \tag{10}$$

where $A_0$ is the drainage area of the orogenic area where the river has its source, outside of the domain defined by $x \in [0, L]$. Assuming that $p = 2$ leads to $k$ being dimensionless.

The right-hand side boundary, at $x = L$, is assumed to corresponds to a base level (a large river or an ocean) such that its elevation remains nil through time. This yields the following boundary conditions:

$$\frac{\partial h}{\partial x}(x = 0, t) = \left(\frac{q_0 w}{K_d (A_0 \nu)^{m+1}}\right)^{1/n} \text{ and } h(x = L, t) = 0 \tag{11}$$





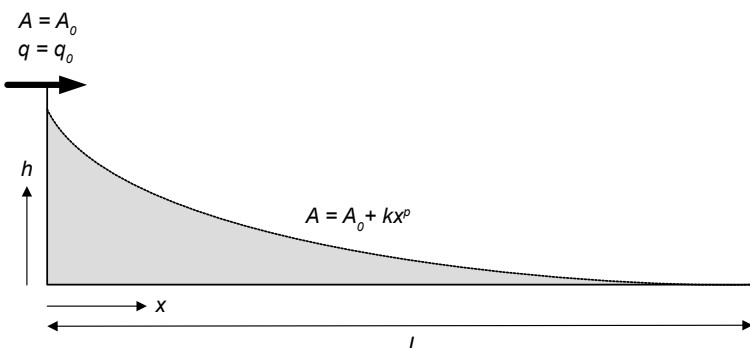

**Figure 1.** Experimental setup.

for the *TL* equation and:

$$h(x = L, t) = 0 \qquad (12)$$

for the $\xi - q$ equation.

## 2.3 Numerical method used

We developed simple time-implicit finite difference schemes to solve these equations numerically under the simplifying assumption that $n = 1$ (see Appendix A for details).

## 3 Results

### 3.1 Steady-state solution

Both equations share the same steady-state solution. Indeed setting $\frac{\partial h}{\partial t} = 0$ and $\nu = 1$ in Equations 3 and 7, we obtain:

$$q(x, t = \infty) = q_0 = \frac{K_d}{w^{m+1}} (A)^{m+1} |S|^n \qquad (13)$$

for the *TL* equation and:

$$q(x, t = \infty) = q_0 = \frac{K_f}{G w^{m+1}} (A)^{m+1} |S|^n \qquad (14)$$

for the $\xi - q$ equation, which yields the following expressions for the topographic elevation:

$$h(x, t = \infty) = \int\limits_x^L \frac{\partial h}{\partial x} \, dx = \int\limits_x^L \left( \frac{q_0 w^{m+1}}{K_d ((A_0 + k x^p))^{m+1}} \right)^{1/n} \, dx$$

Earth **Surface**
**Dynamics**
Discussions

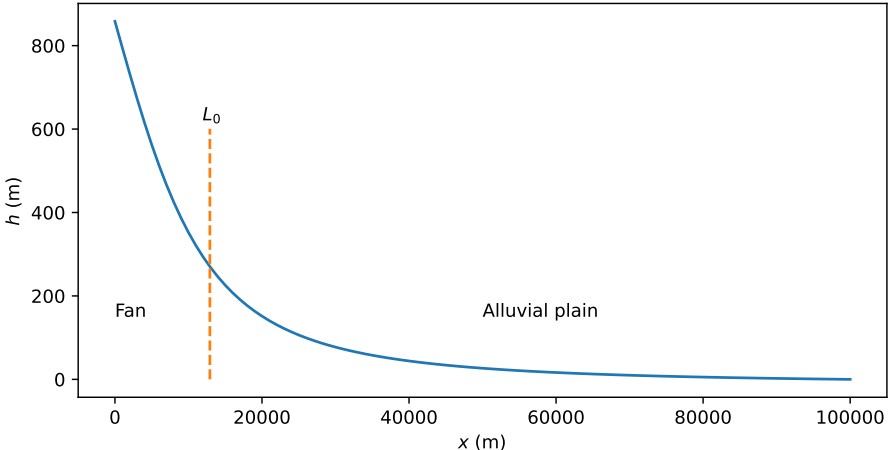

**Figure 2.** Steady-state depositional profile obtained by solving both the $\xi - q$ and *TL* equations using $K_f = 10^{-5}$ m$^{1-2m}$ yr$^{-1}$, $G = 1$ yr m$-1$, $K_d = 10^{-5}$ m$^{-2m}$yr$^{-1}$, $w$=1e4, $m = 0.4$, $n = 1$, $L = 100$ km, $A_0 = 10^8$ m$^2$, $k = 0.6$, $p = 2$ and $q_0 = 10$ m yr$^{-1}$.

$$= \Big( \frac{q_0 w^{m+1}}{K_d(A_0)^{m+1}} \Big)^{1/n} \Big( L \,_2F_1(\frac{1}{p}; \frac{m+1}{n}; 1+\frac{1}{p}; -\frac{kL^p}{A_0}) - x \,_2F_1(\frac{1}{p}; \frac{m+1}{n}; 1+\frac{1}{p}; -\frac{kx^p}{A_0}) \Big) \tag{15}$$

for the *TL* equation and

$$h(x, t = \infty) = \int_x^L \frac{\partial h}{\partial x}\, dx = \int_x^L \Big( \frac{q_0 G w^{m+1}}{K_f((A_0 + kx^p))^{m+1}} \Big)^{1/n} dx$$

$$= \Big( \frac{q_0 G w^{m+1}}{K_f(A_0)^{m+1}} \Big)^{1/n} \Big( L \,_2F_1(\frac{1}{p}; \frac{m+1}{n}; 1+\frac{1}{p}; -\frac{kL^p}{A_0}) - x \,_2F_1(\frac{1}{p}; \frac{m+1}{n}; 1+\frac{1}{p}; -\frac{kx^p}{A_0}) \Big) \tag{16}$$

for the $\xi - q$ equation. $_2F_1(a; b; c; x)$ is the hypergeometric function.

        The two equations have steady-state solutions that have the same form and are identical if/when $GK_d = K_f$. This solution is
shown in Figure 2 for parameter values given in the caption. Its shape is determined by the ratio $kL^p/A_0$ or $L/L_0$ where $L_0 = (A_0/k)^{1/p}$ is the linear size of the upstream catchment or orogenic area. In Figure 3, we show three solutions corresponding to three different values of $L/L_o$. In systems where the size of the depositional area is smaller than or equal to the size of the orogenic area ($L \leq L_0$), the depositional profile is linear (Figure 3a and b). In the more general case where $L > L_0$, the profile is made of two separate sections: in the section near the orogenic area defined by $x < L_0$, the depositional profile is linear
while in the other section defined by $x > L_0$, the profile is curved and progressively tappers towards base level (Figure 3c).

        This geometry is similar to what is observed in natural systems (Bull, 1977; Blair and McPherson, 2009; Bowman, 2019): in the most common situation where the depositional system is much longer than the orogenic system, i.e., $L >> L_0$, the depositional system comprises a steep and constant slope fan, which connects to a much gentler slope alluvial plain; in cases where the depositional system is shorter than the orogenic system, such as next to a mountain neighboring an ocean, the
depositional system is limited to a steep, linear (conic in two dimensions) fan. From here on, we take the convention to name





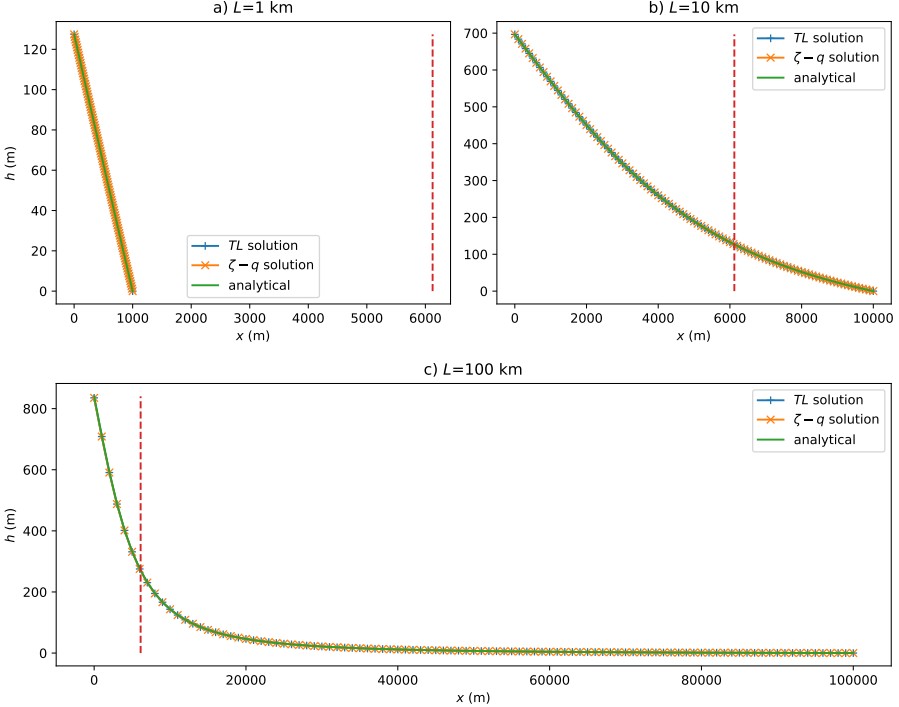

**Figure 3.** Steady-state depositional profiles obtained by solving both the $\xi - q$ and *TL* equations using three different values for the ratio $r = kL^p/A_0$, all other parameters identical to values used for the profile in Figure 2. The dashed line represents the position of the length scale $L_0 = (A_0/k)^{1/p}$. Three curves are shown corresponding to the analytical solution as described by Equation 15 or 16, and the two solutions obtained using the numerical methods described above.

the systems where $L < L_0$ "constrained" systems, i.e., their short length relative to the length of the upstream orogenic area prevents them from building an alluvial plain, whereas those where $L > L_0$ will be called "open" systems, i.e., as they are able to develop an alluvial plain at the foot of their fan.

We note that the parameters $q_0$, $K_f$, $G$, $w$, $K_d$ and $A_0$ controls the height of the depositional system but that its shape, i.e.,
where it transitions from a linear segment to a curve segment, only depends on the ratio of the depositional system size to the orogenic system size (length or area) $A/A_0 = kL^p/A_0$.

The slope of the steady-state solution is given by:

$$S^\infty = -(\frac{w^{m+1}}{K_d} \frac{q_0}{((A_0 + kx^p))^{m+1}})^{1/n} \tag{17}$$

for the *TL* equation and:

$$S^\infty = -(\frac{Gw^{m+1}}{K_f} \frac{q_0}{((A_0 + kx^p))^{m+1}})^{1/n} \tag{18}$$

for the $\xi - q$ equation.





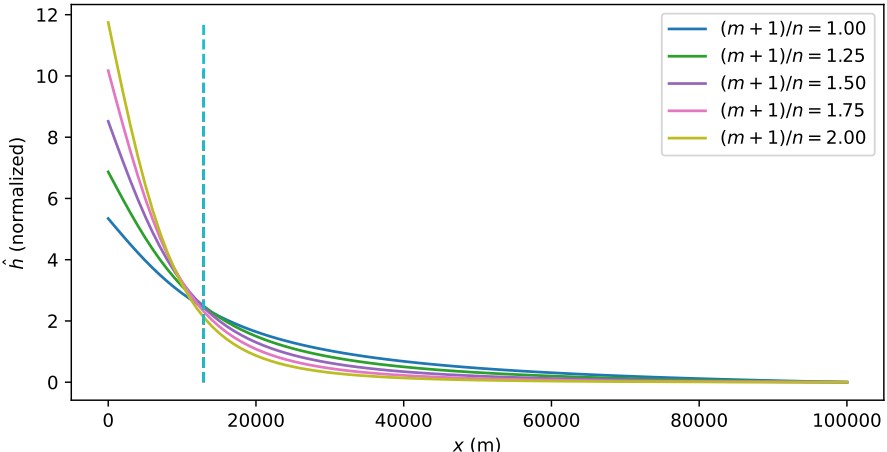

**Figure 4.** Steady-state depositional profiles of open systems obtained by varying the ratio $(m+1)/n$. All other parameters have the same value as in Figure 2. The profile elevations have been scaled so that they have the same mean.

The predicted steady-state slope of the fan system, i.e., at $x = 0$, and alluvial plain, i.e., at $x = L$, are given by:

$$S_0^\infty = -(\frac{w^{m+1}}{K_d}\frac{q_0}{(A_0)^{m+1}})^{1/n} \text{ and } S_L^\infty = -(\frac{w^{m+1}}{K_d}\frac{q_0}{(A_0 + kL^p)^{m+1}})^{1/n} \tag{19}$$

respectively, for the *TL* equation and:

$$S_0^\infty = -(\frac{Gw^{m+1}}{K_f}\frac{q_0}{(A_0)^{m+1}})^{1/n} \text{ and } S_L^\infty = -(\frac{Gw^{m+1}}{K_f}\frac{q_0}{(A_0 + kL^p)^{m+1}})^{1/n} \tag{20}$$

for the $\xi - q$ equation.

For open systems, the ratio $(m+1)/n$ controls the partitioning of the sediment flux between the fan and the alluvial plain. It also controls the difference in slope between the fan and the alluvial plain. For large values of $(m+1)/n$, the fan is much steeper than the alluvial plain and traps a greater proportion of the sediment, for small values of $(m+1)/n$, the fan slope tends towards the alluvial plain slope and a greater proportion of the sediment is deposited in the alluvial plain, as shown in Figure 4.

### 3.2 Transient behavior

We now use the numerical algorithms described in the appendix to investigate the transient behavior of the solution. We first tested that the numerical models yield the steady-state analytical solutions. The results are shown in Figure 3 where the numerical solutions have been superimposed on the analytical solution.

The transient behavior of the solutions to the two equations is shown in Figure 5 for the three situations where $L = L_0/10 < L_0$ (constrained systems, Figure 5a), $L = L_0$ (Figure 5b) and $L = 10L_0 > L_0$ (open systems, Figure 5c). In Figure 5, time has been normalized by the e-folding time scale, $\tau$, determined by fitting each time-elevation curve by an exponential function of the form $1 - exp(-t/\tau)$, while height has been normalized by the maximum height reached at the end of the numerical

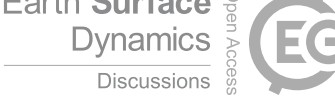

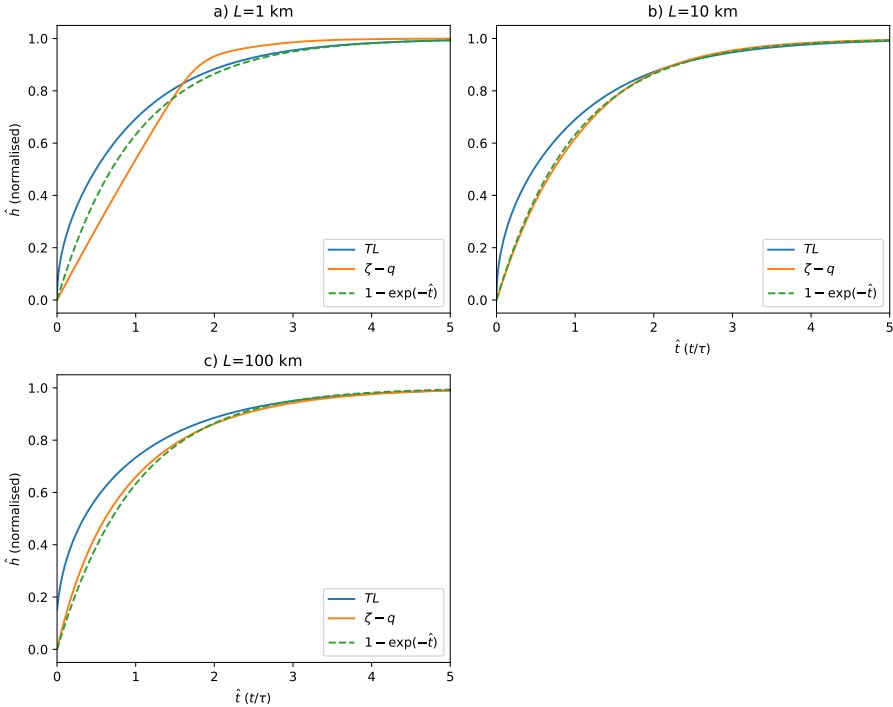

**Figure 5.** Maximum surface elevation as a function of time. Surface elevation is normalized by its maximum value and time by the e-folding time scale, $\tau$. The three panels correspond to different length of the system compared to $L_0$ a) $L = L_0/10$ (constrained systems), b) $L = L_0$ and c) $L = 10L_0$. (open systems) In each panel the curves correspond to the solutions to the $\xi - q$ and $TL$ equations and are compared to the third curve representing an exponential increase of the form $1 - exp(t/\tau)$.

experiment. We see that the time evolution of the solution to the $TL$ equation is always sup-exponential (i.e., it increases faster

than an exponential) but that its shape is independent of whether the system is constrained or open. On the contrary, the shape of the time evolution of the solution to the $\xi - q$ equation is dependent on $L/L_0$, with a more gradual (linear) increase with time for constrained systems and a sup-exponential form for open systems.

To further investigate the transient behavior of the two equations, we show in Figure 6, the evolution of the predicted surface elevation of the system. We show the same information in Figure 7 but after scaling the computed height by the steady-state

height ($h_\infty$) such that we can appreciate the behavior of the solution equally well along the entire profile, even when deposited thicknesses are vey low. We see a major difference between the two equations' behavior. The solution to the $TL$ equation evolves by depositing sediments near the fan apex first until sediments reach the system toe (base level) at which point the solution evolves with a uniform (relative) rate of filling all along its length. The $\xi - q$ equation yields a solution that evolves in the other direction, i.e., from toe to apex, as the sediment fill progresses first towards its steady-state solution near the toe

of the system and then propagates backwards to reach a situation where the relative rate of filling is more uniform over the



Earth **Surface**
**Dynamics**
Discussions

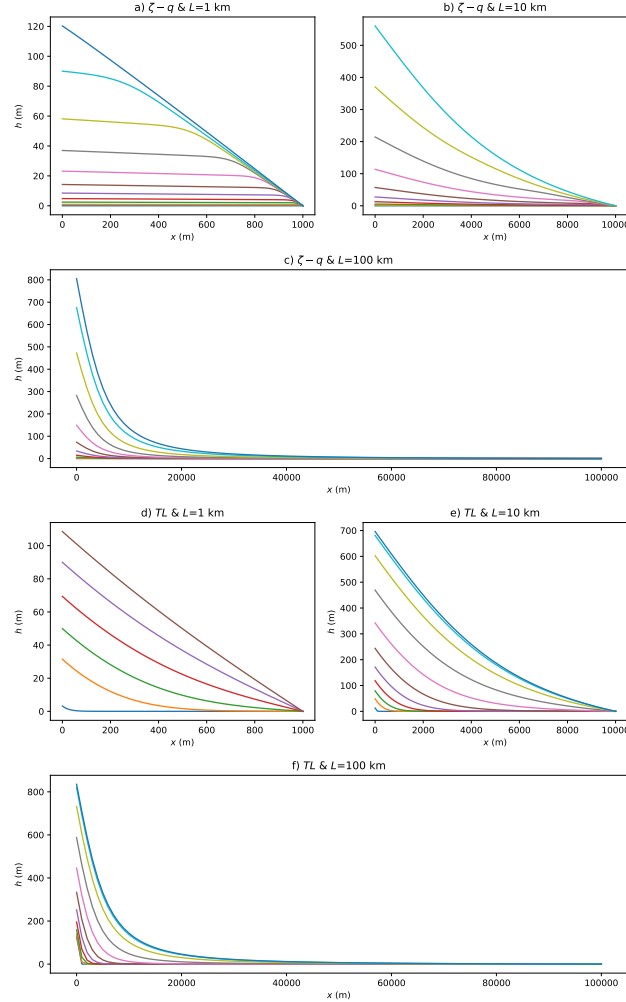

**Figure 6.** Surface elevation at a series of logarithmically spaced time steps obtained by solving the $\xi - q$ and $TL$ equation for different system legnth, $L$, smaller than, equal to or greater than $L_0 = 10$ km.

entire system. This difference in behavior is most striking for constrained systems (i.e. where $L < L_0$), but exists for all system lengths, whether they are constrained or open.

This difference in topographic evolution is accompanied by major differences in the predicted flux out of the system during the transient phase of fan+alluvial plain build up as illustrated in Figure 8 (expressions used to compute the flux values are given in Appendix D). In the $\xi - q$ model, the flux out of the system is instantly finite, i.e., as soon as the sedimentary system starts to grow. In the $TL$ model, the initial flux out of the system is always nil and remains so until the propagation of the sedimentary wedge reaches the toe of the system. In other words the $\xi - q$ model predicts an instantaneous flux response, regardless of the size or character of the system, whereas the $TL$ model predicts a lagged response, with a time lag that appears



Earth **Surface**
**Dynamics**
Discussions

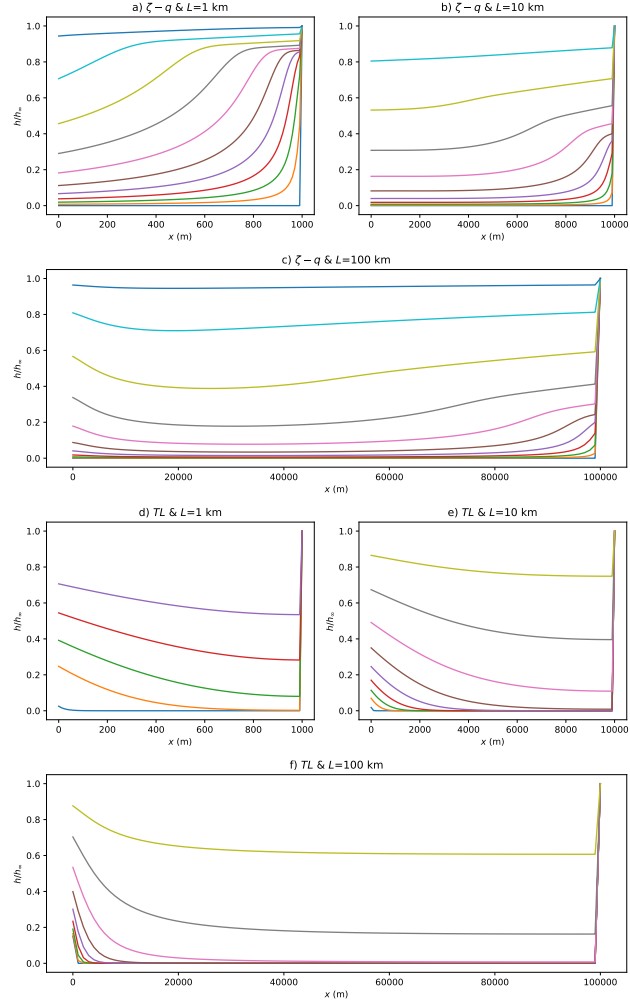

**Figure 7.** Same information as in Figure 6 but using the relative surface topography, i.e., scaled by its steady-state value.

proportional to the length of the system. At all times (scaled by the response time of the system), the outgoing flux predicted by
the $\xi - q$ model is much greater than that predicted by the $TL$ mode. This implies that the $\xi - q$ solution is always much more
'leaky' than the $TL$ solution as it requires a much greater amount of material to transit through the system before it reaches
steady-state.

### 3.3 Response to a step change in incoming sedimentary flux and precipitation rate

We performed a series of experiments in which we abruptly changed the incoming sedimentary flux, $q_0$, or the relative pre-
cipitation rate $\nu$. The results are shown for an increase in sediment flux in Figure 9, and in the Supplementary Material for a





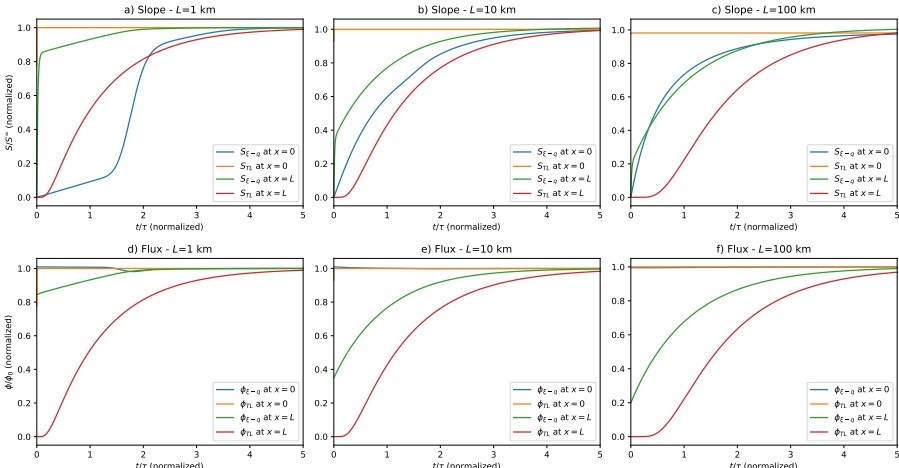

**Figure 8.** Evolution of the slope (top panels) and flux (bottom panels) normalized by their steady-state values at both ends of the system as a function of time.

decrease in sediment flux (Figure S1), for an increase in relative precipitation rate (Figure S2) and for a decrease in relative precipitation rate (Figure S3, for both models.

We see that for an increase in sedimentary flux (Figure 9), the system moves back towards a new steady-state profile first near the toe of the system for the $\xi - q$ equation and first near the apex of the fan for the $TL$ equation. In the $\xi - q$ solution,

the aggradation process initiates simultaneously at all distance along the system profile, whereas in the $TL$ solution, the aggradation starts near the apex of the fan but very rapidly propagates throughout the entire system length. The differences between the two solutions is only substantial if one compares the evolution of the relative thickness and are more marked for constrained fan systems.

### 3.4 Response time

We have shown that an e-folding time scale, $\tau$ can be derived from the shape of the evolution equation of the maximum surface elevation of the sedimentary system. This time scale is called to *response time* of the system as it corresponds to the time it takes for the system to reach its steady-state shape but, more generally, the time it takes for the system shape to respond to change in its external forcings (incoming sediment flux or precipitation rate).

In Figure 10, we show the results of 24 numerical experiments in which we solved the $TL$ and $\xi - q$ equations varying the

value of $L$. For each experiment, we computed the response time by fitting an exponential curve of the type $1 - \exp(-t/\tau)$ to the computed evolution of maximum elevation with time (upper panels in Figure 10). The $\xi - q$ response times are reported in Figure 10c and the $TL$ response time are reported in Figure 10d as 24 circles. We see that for constrained systems ($L < L_0$), the $TL$ response time varies quadratically with $L$, whereas the $\xi - q$ response time varies linearly with $L$. However, this dependence changes dramatically for open systems, i.e., when $L$, becomes greater than the size of the orogenic system, $L_0$. This threshold



Earth **Surface**
**Dynamics**
Discussions

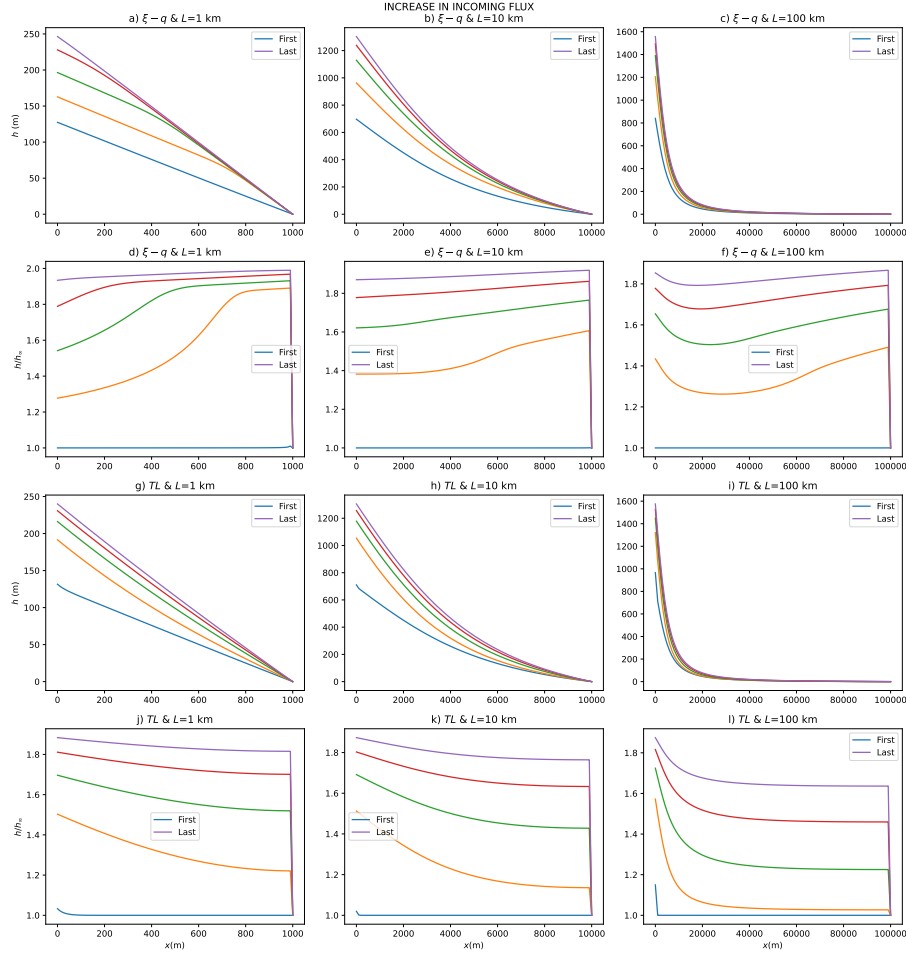

**Figure 9.** Evolution of the surface topography following an increase in incoming sediment flux by a factor of 2. $\xi - q$ solution in the top 6 panels and $TL$ solution in the bottom 6 panels. Panels d) to f) contain the same information as panels a) to c) but using the topographic elevation normalized by its final, steady-state value. Idem for panels j) to l) with respect to panels g) to i).

is marked by a star in both panels of Figure 10. For intermediate size systems, i.e., when $L_0 < L < 100L_0$, there is almost no dependence of either response times on $L$. For large open systems, i.e., when $L >> L_0$, the $\xi - q$ response time varies as $L^{1-mp}$ while the $TL$ response time varies as $L^{2-(m+1)p}$.

To understand this behavior, we go back to Equations 3 and 7, to derive scaling relationships for the $TL$ and $\xi - q$ response times, $\tau_{TL}$ and $\tau_{\xi-q}$, respecitively. For the $TL$ equation, the scaling gives:

$$\frac{h_0}{\tau_{TL}} = \frac{K_d}{L}(\frac{A}{w})^{m+1}(\frac{h_0}{L})^n \tag{21}$$





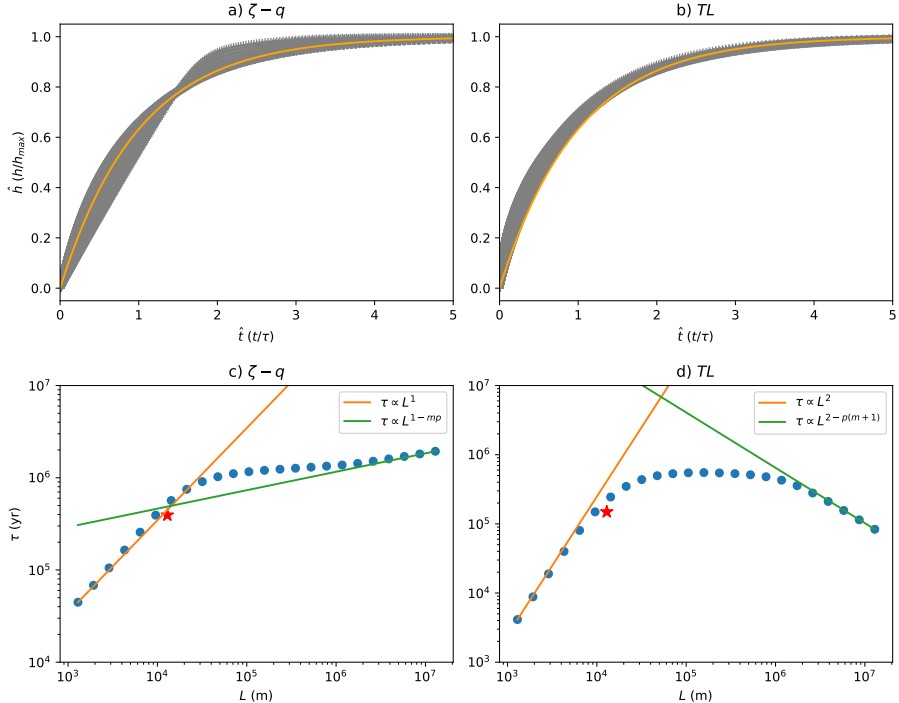

**Figure 10.** Computed response times for 24 numerical experiments in which the length of the model, $L$, was varied. a) and b) time evolution of the maximum height of the depositional system for all 24 experiments (grey curves) on which the exponential fit has been superimposed (orange curve). c) and d) corresponding response time estimates (blue circles) on which lines describing the asymptotic behaviors discussed in the text have been superimposed. Note that the absolute values of the response times should be considered with caution as they correspond to a specific choice of relatively poorly constrained values of the rate parameters, $K_f$ and $K_d$.

From the steady-state solution (Equation 15), we know that

$$h_0 = (\frac{q_0}{K_d})^{1/n}(\frac{w}{A_0})^{(m+1)/n}L \tag{22}$$

which gives:

$$\tau_{TL} = L^2 q_0^{1/n-1} K_d^{-1/n} w^{(m+1)/n} A_0^{-(m+1)/n} \quad \text{when } L \leq L_0$$

$$\tau_{TL} = L^{2-p(m+1)} q_0^{1/n-1} K_d^{-1/n} w^{(m+1)/n} A_0^{(m+1)(n-1)/n} k^{-(m+1)} \quad \text{when } L > L_0 \tag{23}$$

For the $\xi - q$ equation, the scaling goes as:

$$\frac{h_0}{\tau_{\xi-q}} \propto K_f(\frac{A}{w})^m(\frac{h_0}{L})^n + \frac{Gw}{A}(q_0 - \frac{h_0}{\tau_{\xi-q}}L) \tag{24}$$

with:

$$h_0 \propto (\frac{q_0 G}{K_f})^{1/n}(\frac{w}{A_0})^{(m+1)/n}L \tag{25}$$



Two cases must be considered, depending on the value of the dimensionless number:

$$\delta = \frac{LGw}{A} = \frac{LGw}{(A_0 + kL^p)} \tag{26}$$

If the equation is dominated by the erosional term ($\delta < 1$), the scaling goes as:

$$\tau_{\xi-q} = L^1 q_0^{1/n-1} K_f^{-1/n} G^{1/n-1} w^{(m+1)/n-1} A_0^{1-(m+1)/n} \text{ when } L \leq L_0$$

$$\tau_{\xi-q} = L^{1-mp} q_0^{1/n-1} K_f^{-1/n} G^{1/n-1} w^{(m+1)/n-1} A_0^{-(m+1)(n-1)/n} k^{-m} \text{ when } L > L_0 \tag{27}$$

whereas if the equation is dominated by the depositional term ($\delta > 1$), the scaling goes as:

$$\tau_{\xi-q} = L^2 q_0^{1/n-1} K_f^{-1/n} G^{1/n} w^{(m+1)/n} A_0^{-(m+1)/n} \tag{28}$$

regardless of the value of $L$ with respect to $L_0$, which is the same scaling as that of the $TL$ equation for $L < L_0$ and $n = 1$.

Interestingly, $\delta$ is a non-linear function of $L$ that reaches a maximum value:

$$\delta_{max} = \frac{L_0^{1-p} Gw}{k} \frac{(p-1)^{1-1/p}}{p} \tag{29}$$

for $L = L_0(p-1)^{-1/p}$. For $p = 2$, $\delta$ is maximum for $L = L_0$.

We see that for constrained systems, the $TL$ response time scales as the $n+1$st power of length but that, for open systems, this scaling is inverted, i.e., the $TL$ response time decreases with length, almost regardless of the linearity of the system. For constrained systems, the $\xi - q$ response time scales at most with the length of the system but for open systems, the scaling drops to a small power. Again this behavior is relatively independent of the linearity of the system.

Both response times are independent of the incoming flux, $q_0$, in linear systems and decrease with a small power of $q_0$ in non-linear systems. The $TL$ response time is always inversely proportional to the precipitation rate to a power close to unity, whereas the $\xi - q$ response time is only weakly dependent on precipitation rate in erosion-dominated cases ($\delta < 1$) and inversely proportional to precipitation rate in deposition-dominated systems. Both time scales vary inversely with the rate constants (diffusivity or erodibility) and, in the linear case, the $\xi - q$ response time is independent of $G$ in erosion-dominated

systems and increases linearly with $G$ in deposition-dominated systems.

In Appendix B, we show how the response time scales with the various characteristics of the systems for a range of values of the exponents $m$ and $n$.

In Appendix C, we present the results of several series of numerical experiments demonstrating the validity of the scaling we present above.

## 3.5    Comparison of response time scales

We have seen that the two equations lead to an identical steady-state solution when the model parameters are judiciously chosen to be in the ratio $GK_d = K_f$. For their transient behavior to be similar requires (at minima) that their response times be also similar. This implies for constrained systems that:

$$\frac{\tau_{TL}}{\tau_{\xi-q}} = \frac{L^2 q_0^{1/n-1} K_d'^{-1/n} w^{1/n} A_0^{-(m+1)/n}}{L^1 q_0^{1/n-1} K_f^{-1/n} G^{1/n-1} w^{1/n-1} A_0^{1-(m+1)/n}} = \frac{LGw}{A_0} = 1 \tag{30}$$



Earth **Surface**
**Dynamics**
Discussions

and for open systems that:

$$\frac{\tau_{TL}}{\tau_{\xi-q}} = \frac{L^{2-p(m+1)}q_0^{1/n-1}K_d'^{-1/n}w^{1/n}A_0^{(m+1)(n-1)/n}k^-(m+1)}{L^{1-mp}q_0^{1/n-1}K_f^{-1/n}G^{1/n-1}w^{1/n-1}A_0^{(m+1)(n-1)/n}k^{-m}} = \frac{L^{1-p}Gw}{k} = 1 \tag{31}$$

For the solution to the two equations to have the same transient behavior, regardless of the length of the system, we must have:

$$\frac{L^2}{A_0} = \frac{L^{1-p}}{k} \text{ or } L = L_0 \tag{32}$$

It is therefore impossible for both equations to reproduce the transient behavior of constrained AND open systems with a

unique set of parameters; only the particular case of $L = L_0$ can.

Considering now a system of arbitrary length $L$, the ratio of the two times scales is:

$$\frac{\tau_{TL}}{\tau_{\xi-q}} = \frac{LGw}{\max(A_0, A)} \tag{33}$$

showing that, for values of $G$ close to unity, and for a choice of model parameters that lead to the same steady-state solution, the $\xi - q$ model will generate longer time scales than the $TL$ model in a ratio equal to the ratio of the total upstream drainage

area to the area of the flood-plain (the part of the drainage area where active sedimentation/erosion and transport takes place).

### 3.6  Periodic variations in input flux

We now investigate how the system reacts to a periodic perturbation in incoming sedimentary flux from the source or orogenic area. We will consider first how the system shape reacts and then how it transmits the sedimentary flux signal from the source (the orogenic system boundary) to the sink (the base level boundary).

In Figure 11a and b, we show the gain, $\Gamma_h$, and phase, $\phi_h$ of the response of the system measured as the variation of the maximum topography, i.e., near the orogenic front of the sedimentary system, as a function of the forcing period normalized by the response time. The gain is the ratio of the relative amplitude of the response (i.e., the amplitude of the variations in maximum height scaled by the maximum height at steady-state) to the relative amplitude of the forcing (i.e., the amplitude of the incoming flux variations scaled by the mean incoming flux). The phase is the phase shift between the response and the

forcing normalized by the forcing period. A phase shift of 0.25 corresponds to an angular phase shift of $90°$.

We see that for both models, the gain decreases from 1 to 0 as the forcing period decreases. For rapid (or short) forcing periods, i.e., much smaller than the characteristics response time, the gain tends towards 0, while for slow (or long) forcing periods, the gain tends towards 1. In other words, the system shape is not affected by variations in incoming flux that are smaller (or faster) than the characteristic time scale, regardless of whether the system is constrained or open, while variations

in sedimentary flux are fully expressed as variations in deposited sediment thickness when the variations in incoming flux are longer than the characteristic time scales, regardless also of whether the system is constrained or open.

We also see that the phase shift is a strong function of the forcing period: for large forcing periods, the phase shift tends toward 0, while for forcing periods that are equal to or smaller than the characteristic time scale, it grows to reach values of about 0.125 (or $\pi/4$) for the $TL$ model and 0.25 (or $\pi/2$) for the $\xi - q$ model, regardless of whether the system is constrained

or open.





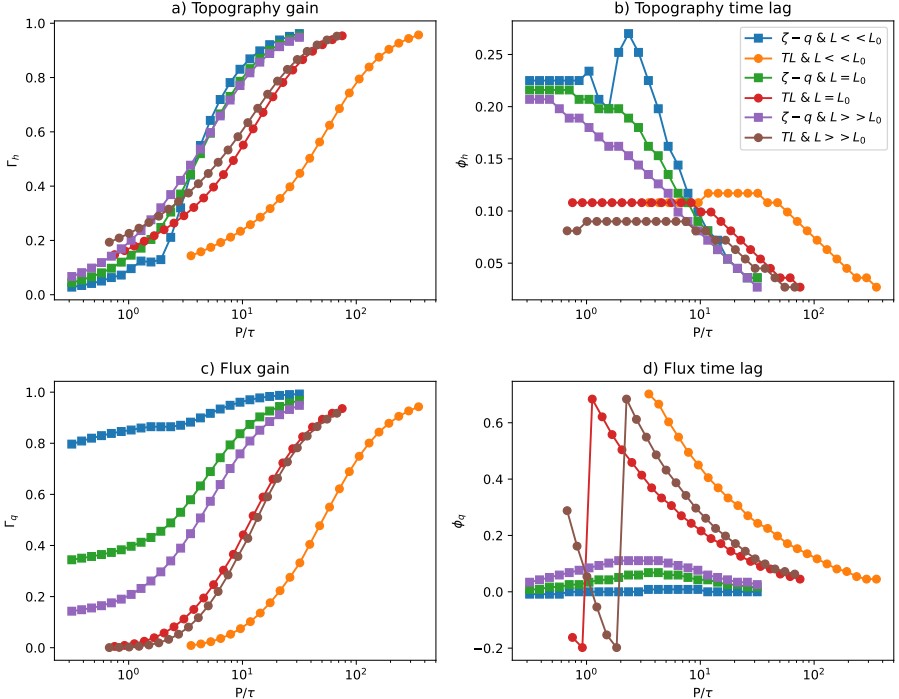

**Figure 11.** Computed a) gain, $\Gamma_h$, and b) phase, $\phi_h$, of the response of the system shape as a function of the period of the imposed periodic incoming flux normalized by the system's response time for constrained ($L < L_0$), intermediary ($L = L_0$) and open ($L > L_0$) systems using the $\xi - q$ and $TL$ models. Computed c) gain, $\Gamma_q$, and d) phase, $\phi_q$, of the outgoing sedimentary flux.

In summary, variations in system morphology or height will be recorded in the sedimentary record as variations in deposited (and eroded) sediment thickness. These will be largest near the orogenic front but will be noticed at all locations within the sedimentary system. At most (i.e., when $\Gamma_h = 1$) their amplitude will be directly proportional to the amplitude of the flux variations. When the system most strongly reacts to the variations in incoming flux (i.e., when $\Gamma_h \approx 1$), it does it in phase

with the forcing ($\phi \approx 0$); phase lags only appear when the response is weak. This means that if a system is responding in a noticeable manner to a change in incoming sedimentary flux, it does it with a minimal time lag.

In Figure 11c and d, we show the gain, $\Gamma_q$, and the time lag, $\phi_q$, between the incoming and outgoing fluxes, i.e., computed at the right hand side boundary of the model or its base level. These quantities characterize the ability of the system to transmit sedimentary flux signals across their length.

Interestingly, the gain functions are radically different for the $\xi - q$ and the $TL$ models. Regardless of whether the system is constrained or open, the $TL$ model predicts that the gain varies from 1 to 0 as the forcing period decreases from values larger than to values lower than the characteristic time scales. The $TL$ model predicts that a sedimentary system can only propagate signals that vary slower than their characteristic time scales. Note also that as the signal is damped (with decreasing forcing



Earth **Surface**
**Dynamics**
Discussions

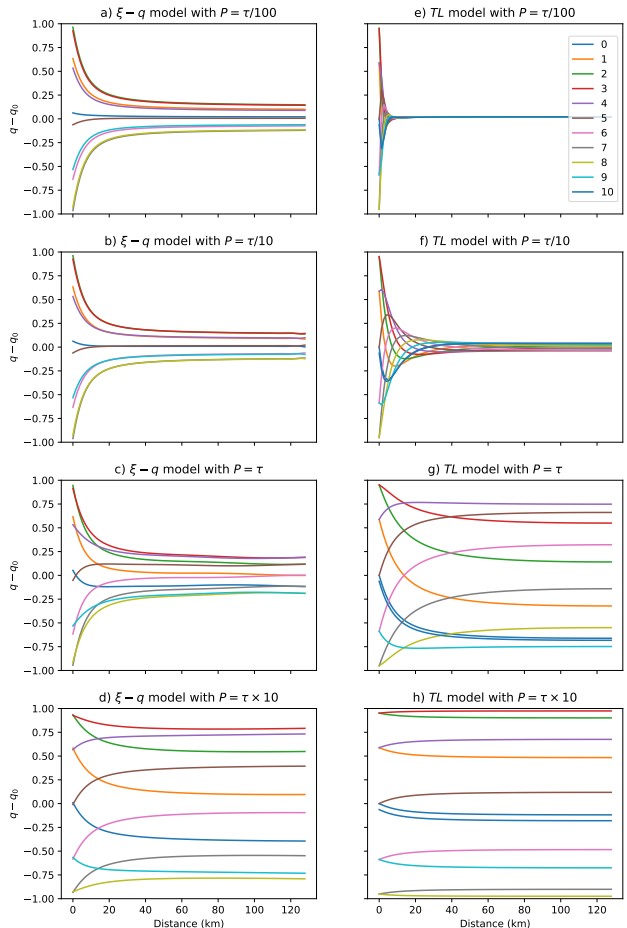

**Figure 12.** Computed flux profiles across the sedimentary system at ten time steps within one of the imposed incoming flux cycle; a) to d) using the $\xi - q$ model and e) to h) using the $TL$ model. Going from left to right, the forcing period is equal to $\tau/100$, $\tau/10$, $\tau$ and $10 \times \tau$, respectively. In all cases shown $L = 10L_0$.

period) the phase shift increases to become more than a quarter cycle out of phase ($\phi > 0.25$) with the input signal. This is

because the $TL$ model predicts that incoming flux variations propagate as standing waves across the sedimentary system.

This is illustrated in Figure 12, where we show the computed sedimentary flux across the entire system for ten equally spaced time steps within a forcing period (see Appendix D for the expressions used to compute the fluxes for both models). We see that, for the $TL$ model, the slow signals are transmitted through the entire system whereas rapid signals are not. In the situation where the forcing period is similar to the characteristic response time (panel e in Figure 12), one see a standing wave pattern

developing across the system. This is because in the $TL$ model, any signal must be transmitted by changes in slope and such change can only occur over a time equal to the characteristic time scale.



Earth **Surface**
**Dynamics**
Discussions

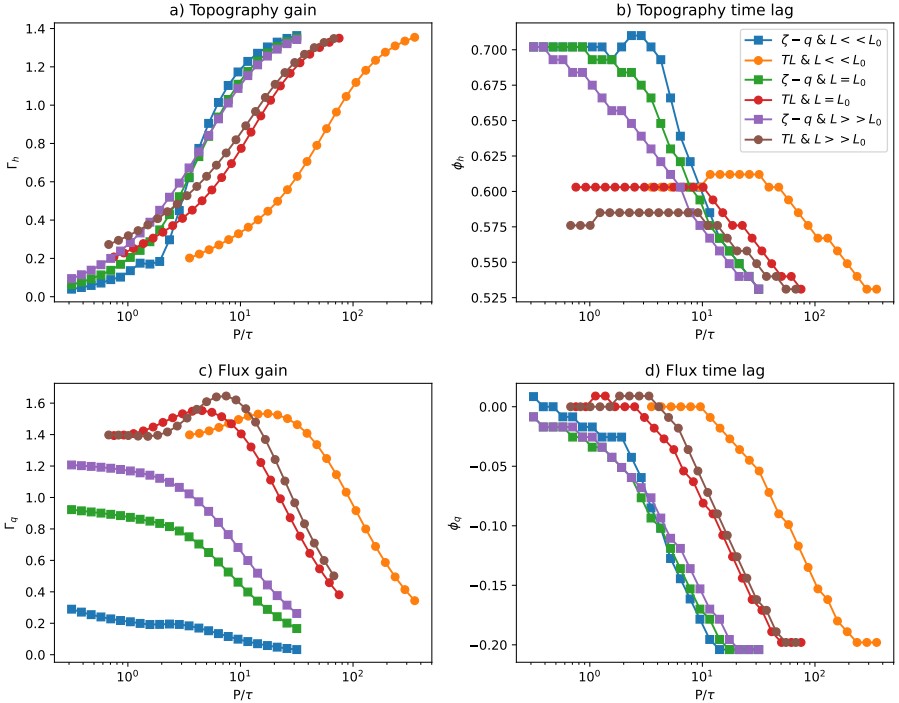

**Figure 13.** Computed a) gain and b) phase of the response of the system shape as a function of the period of the imposed periodic precipitation rate normalized by the system's response time for constrained ($L < L_0$), intermediary ($L = L_0$) and open ($L > L_0$) systems using the $\xi - q$ and $TL$ models. Computed c) gain and d) phase of the outgoing sedimentary flux.

To the contrary, we see in Figures 11c and d and Figure 12 that using the $\xi - q$ model the sedimentary system is predicted to transmit information along its entire length without much change in slope/shape. As stated by Davy and Lague (2009), and contrary to previous under-capacity formulations such as that of Kooi and Beaumont (1994), the $\xi - q$ model predicts that the system is uniformly under-capacity, i.e., along its entire length. It does not display a transition from detachment limited near the source to transport limited near the base level. Thus, it is able to transmit signals nearly instantaneously and with much less sensitivity to the forcing period. This is seen in Figure 11c where the flux gain function never reaches 0 even for very rapid forcing periods. This is further illustrated in Figure 12a to d, where the incoming flux variations are transmitted throughout the entire length of the system even if the forcing period is much shorter than the characteristic time of the system (Figure 12a).

### 3.7 Periodic variations in precipitation rate

We performed a series of numerical experiments in which we varied the precipitation rate, $\nu$, in a sinusoidal fashion, for a range of periods encompassing the response time of the sedimentary system. The results are shown in Figure 13 and are relatively similar for the $\xi - q$ and $TL$ models.





They show that variations in precipitation rate cause variations in deposited thickness in the sedimentary system that vary in

amplitude as a function of the forcing period, similarly to variations in shape/thickness predicted for a incoming sedimentary flux forcing: for forcing periods that are smaller than the response time of the system, the amplitude tends towards zero and increases with the length of the forcing period. However, predicted gain values for very long forcing periods ($> 10$ to $100 \times \tau$) tend towards $1.6 > 1$. This is because the relative precipitation rate comes to the power $1 + m = 1.6$ in the amplitude of the analytical solutions (Equations 15 and 16).

Another major difference is that the shape response is in complete phase opposition ($\phi = 0.5$) for the largest gain values (corresponding to long forcing periods) and increases to even greater phase values for forcing period smaller than the response time. This is because the relative precipitation rate appears in the denominator of the amplitude of the analytical solutions; in other words, the thickness of the sedimentary deposit is inversely proportional to the relative precipitation rate (to the power $m + 1$).

The outgoing flux gain and time lag are shown in panels c and d of Figure 13. Interestingly, the gain values decrease with increasing periods. This is because for precipitation rate forcing periods that are larger than the characteristic time scale, the depositional system is able to adapt its shape to transport the incoming flux at all times, regardless of its transport capacity (determined by the precipitation rate). As for the topographic gain, values can be larger than one (up to $m + 1 = 1.6$). The lag is nil for large values of the gain and reaches a quarter period for small values of the gain (corresponding to long periods).

We further illustrate this point by showing in Figure 14 values of the flux across the entire system at ten times during one of the precipitation rate cycles. The pattern is inverted compare to that observed for a cycling forcing in incoming sedimentary flux (i.e. compared to results shown in Figure 12): fast varying perturbations are transmitted or even amplified whereas slow varying perturbations are completely damped, for both the $\xi - q$ and $TL$ models.

## 4    Discussion

### 345    4.1    New analytical solution

We have derived a new analytical solution for the shape of a sedimentary system comprising a fan/piedmont deposit and the adjacent alluvial plain. This analytical solution shows that both model formulations can reproduce these first-order features and that, in both models, the transition between fan and plain deposits corresponds to the point where the contribution to runoff from the sedimentary system equals that of the upstream orogenic area. The fan is steeper, more linear and its size is controlled

by the size of the upstream catchment and the along-stream rate of increase of discharge in the sedimentary system (the exponent of the assumed Hack's law). The sedimentary plain is characterized by a smaller gradient and has a concave profile. The analytical solution also implies that the change in surface gradient between the fan and the plain is a strong function of the ratio $(m + 1)/n$, which must be of the order of unity to reproduce the observed change in surface slope.

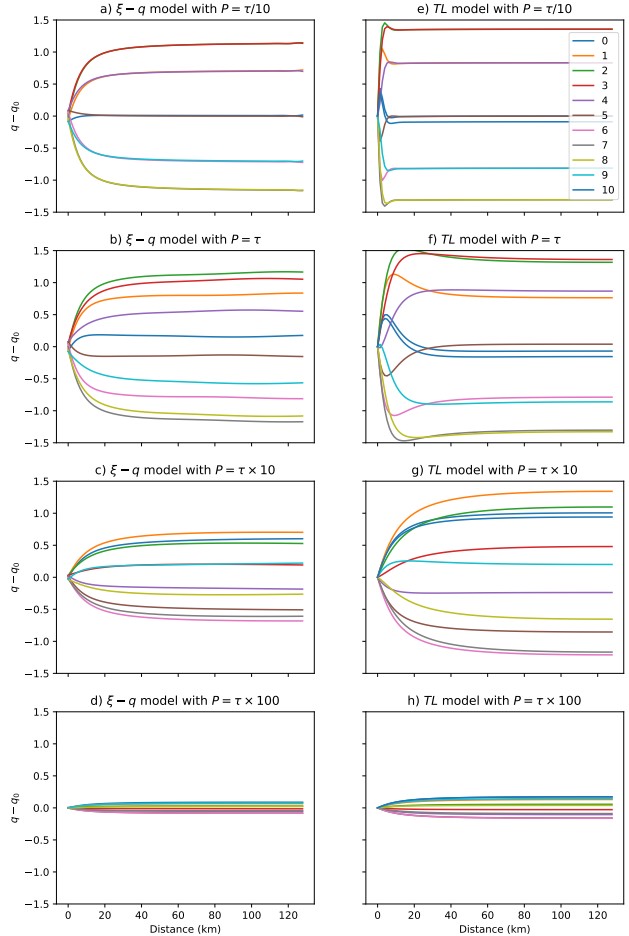

**Figure 14.** Computed flux profiles across the sedimentary system at ten time steps within one of the imposed precipitation rate cycle; a) to d) using the $\xi - q$ model and e) to h) using the $TL$ model. Going from left to right, the forcing period is equal to $\tau/10$, $\tau$, $10\tau$ and $100 \times \tau$, respectively. In all cases shown $L = 10L_0$.

Our new analytical solution explains the globally observed linear relationship between fan area, $A_{fan}$ and upstream/orogenic drainage are, $A_0$ (see Figure 15a from Blair and McPherson (2009)) as:

$$A_{fan} \propto L_0^2 = (\frac{A}{k})^{2/p} \approx \frac{A_0}{k} \tag{34}$$

as well as the inverse relationship between the slope of the fan, $S_{fan}$, and the upstream drainage area (see Figure 15b from (Mouchené et al., 2017)) as:

$$S_{fan} \propto A_0^{-(m+1)/n} \tag{35}$$





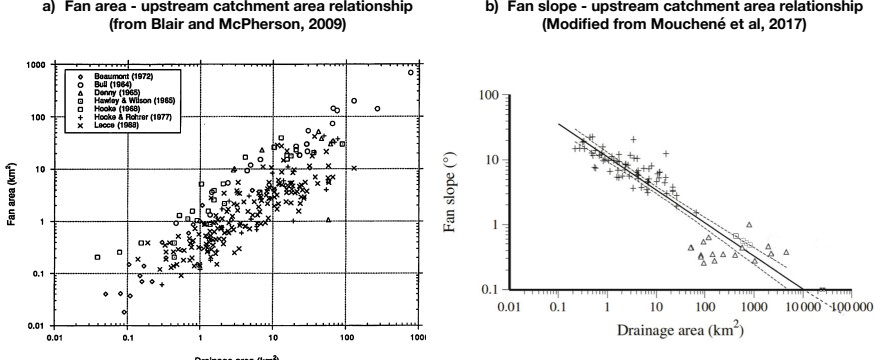

**Figure 15.** a) Linear relationship between fan area and upstream catchment drainage area from a wide range of sources compiled by Blair and McPherson (2009). See Blair and McPherson (2009) for references cited in the figure legend. b) Power law relationship between fan slope and drainage area as compiled by Mouchené et al. (2017).

It also explains the relationship between fan slope and sediment flux scaled by upstream water discharge observed in experimental settings (Whipple et al., 1998) as:

$$S_{fan} \propto \left( \frac{q_0}{(A\nu)^{m+1}} \right)^{1/n} \tag{36}$$

    As observed in nature, the analytical solution also shows why sedimentary systems are constrained in their size and shape by the location of or distance to their base level (Blair and McPherson, 2009). If that distance is small, such as in situations where

a large river, a lake or an ocean is situated in the vicinity of the orogenic front, the fan is steep and almost perfectly linear and connects directly to the base level. This morphology is observed in many small systems such as the Death Valley fans (Bull, 1977; Blair and McPherson, 2009). On the contrary, if that distance is large, the system is open and the fan can develop into its natural size and connects to a lower gradient alluvial plain where a concave long-profile develops to connect the fan to the base level. We wish to stress here that the qualifier "small" and "large" do not refer to the absolute size of the system but must

be considered in comparison to the size of the upstream catchment area.

    Finally, the analytical solution also demonstrates that the shape and size of a fan can reach steady-state values even if the fan does not extend to its base level and can therefore be seen as a simple solution to the so-called "alluvial fan problem" described by Lecce (1990). Our solution also demonstrates why this debate about whether fans reach steady-state sizes/shapes could not be resolved by laboratory scaled experiments as most do not include a contribution to runoff from the depositional area.

## 375   4.2   What do the two models have in common?

The $\xi - q$ and $TL$ models share their steady-state solution. With an appropriate choice of rate parameters, i.e., $K_d$ and $K_f$, and dimensionless constant $G$, the two solutions can be made identical. Acknowledging that we do not know the value of either of these three parameters leads to the conclusion that the two models cannot, in practical terms, be differentiated based on the





shape of their long-term, steady-state solution. As noted above, both models can reproduce the first-order features of natural

sedimentary/by-pass systems, which implies that they should not be discriminated on that basis.

Both models share a similar behavior under a wide range of situations in that their transient response is, in all cases, controlled by the ratio of the period of the forcing to their response time scale. This is however true of most systems controlled by diffusion or advection-type differential equations and is therefore not surprising.

In particular, regardless of which model is used, only slow incoming sedimentary flux variations (i.e., with a period greater

than the response time of the system) will result in measurable variations in deposited/eroded sediment thickness in the sedimentary system, whereas only fast variations in precipitation rate will result in measurable variations in sediment thickness.

### 4.3 Where do the two models differ?

The $\xi - q$ and $TL$ models differ in their transient behavior in three ways. Firstly, they differ by the value of their response time with the $\xi - q$ model characterized by longer response times than the $TL$ model under the assumption that model parameters

are such that the two models predict the same steady-state solution. The ratio of the $\xi - q$ to $TL$ model response times is a function of the ratio of the area under active sedimentation/transport/erosion and the drainage area. The reason for greater response times for the $\xi - q$ model is that the model predicts a transient response that is uniformly distributed along its length, whereas the $TL$ model responds by progressively changing its surface slope across the model. This implies that the $\xi - q$ model predicts that any perturbation is instantaneously propagated to the system base level and affects the outgoing flux through base

level making the system more "leaky" than the $TL$ model. One can show (see Appendix E) that the $TL$ response time for a constrained fan system (i.e., where $L << L_0$) is approximately equal to twice the volume of the fan divided by the incoming flux, which indicates that during the transient build-up of the fan, most of the material introduced into the fan from the orogenic area has been stored into the fan. The $\xi - q$ response time is greater by a factor $\frac{LGw}{A_0}$.

Secondly, they differ by the dependence of their response time scale on the length of the system and, to a lesser degree, on

the size of the upstream area and the width of the flood plain. Constrained systems (or systems that are not able to develop a plain in front of their fan) have a response time that varies as the square of the length of the system in the $TL$ model and as the length of the system only in the $\xi - q$ model. Both models predict a response time that shows a very weak dependence on the length of the system for intermediary systems but, for very long systems, the $TL$ model response time varies inversely with the length of the system (the longer the system, the shorter the time scale), whereas the $\xi - q$ model response time increases with

the system length.

Thirdly, the models differ by the way they are able to transmit sedimentary signals. According to the $TL$ model, only slow perturbations in incoming sedimentary flux will be transmitted through the system and therefore likely to be recorded in the adjacent offshore basin. If one uses the $\xi - q$ model to represent a sedimentary system, all flux perturbations will be transmitted to the offshore basin, regardless of the rate at which they take place. The higher frequency signals will be slightly damped

compared to the low frequency signals, but all are transmitted in a measurable manner.





### 4.4 Are the differences meaningful?

An important question to address is whether these differences are relevant and/or important and in which context. Considering that both models are reduced-complexity models that should only be used to investigate the large-scale and long-term behavior of a sedimentary system, we suggest that great care should be taken in deciding which of the two models to use to investigate the transient behavior of sedimentary systems and in particular their response to external forcing of tectonic or climatic origin. This is particularly true in so-called source-to-sink studies which aim at inverting the marine sedimentary record to infer the timing and amplitude of tectonic or climatic changes in the source area, i.e., the mountain. We have shown that the so-called 'transfer area' that consists of the onshore sedimentary system that builds up at the base of the mountain (the fan or megafan and the adjacent alluvial plain) would appear to have very different properties whether one uses the $\xi - q$ or $TL$ model to represent it. Most worrying is the fact that according to the $TL$ model some sedimentary signals cannot be transmitted across the transfer one, while the $\xi - q$ model does not predict such a behavior. More fundamentally, that the response time scales predicted by the two models are different and show a different scaling/dependence with regard to system length should also be noted and lead to diametrically opposite conclusions regarding the existence an/or nature of orogenic processes and their preservation in sedimentary systems.

### 4.5 What observations could be used to tell the models apart?

To differentiate between the two models or representations of sedimentary processes, one obviously needs to search into observational constraints during transient periods either in the early stages of development of a sedimentary system or during its response to external perturbations. The first type of observations are not easily made as the early stages of development of a fan are often buried beneath large sedimentary sections. The second type of observations require accurate dating or correlation across opposite parts of the sedimentary system, i.e., near the orogenic front and either at the base of the fan or near the base level of the sedimentary system.

Another test comes from the prediction that, according to the $\xi - q$ model some signals should propagate into the marine sedimentary record even if they are shorter than the response time of the system and, this, regardless of whether such signals leave a stratigraphic record in the continental sedimentary system. In view of the wide range of periods (down to the shortest of Milankovitch periods) that are routinely observed in the marine sedimentary record, one would tent to favor the $\xi - q$ model over the $TL$ model. However, one must exercise caution in drawing such a conclusion as such signals might be the product of variations in sea level rather than variations in sediment flux from the source/orogenic area.

The distribution of grain size in continental sedimentary systems has been used to constrain their transient behavior (Armitage et al., 2011; Duller et al., 2010) but most studies have been based on the approximation that deposition is equal to basin subsidence (Duller et al., 2010) or have used a non-linear diffusion ($TL$) approach (Armitage et al., 2011). It would be, potentially, very informative to perform similar studies using the $\xi - q$ model and note if noticeable differences emerge between the two approaches and whether they are of sufficient amplitude to be discerned in field observations.

Earth **Surface**
**Dynamics**
Discussions

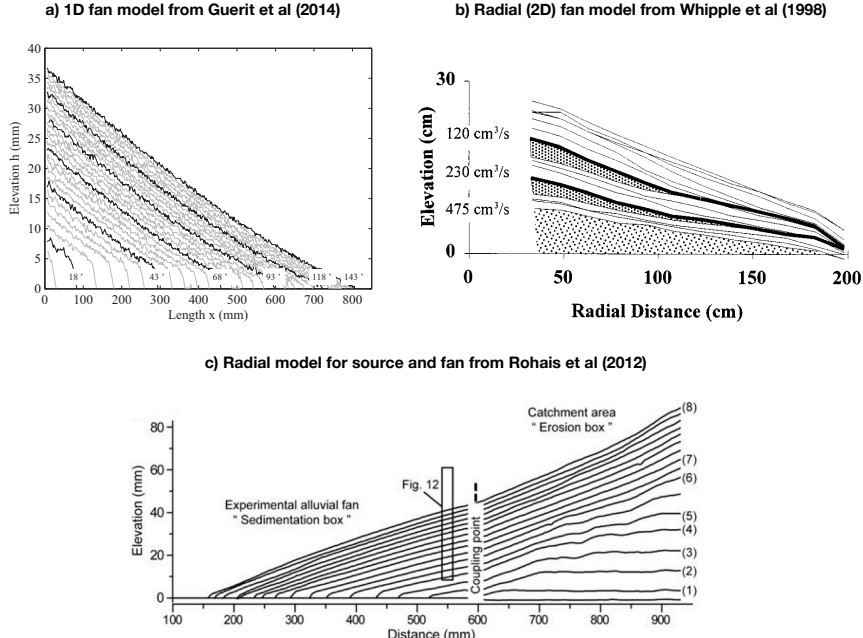

**Figure 16.** Stratigraphy observed in three laboratory experiments displaying a behaviour similar to that predicted by a) and c) the $TL$ model (from Guerit et al. (2014) and Rohais et al. (2012)) and b) the $\xi - q$ model (from Whipple et al. (1998)). Note that none of these experiments included rainfall in the fan area.

Laboratory experiments could be used but one must remember that they only reflect the behavior of scaled-down materials and conditions, not the natural world. Furthermore, looking at the results of several published experiments tend to demonstrate

445 that both behaviors are observed. In Figure 16a, b and c, we show the results of three experiments under relatively similar conditions: the first and third ones from Guerit et al. (2014) and Rohais et al. (2012) show a sedimentary fan developing by propagation of a self-similar system under constant slope, as predicted by the $TL$ model (Figure 6d) whereas the second one from Whipple et al. (1998) shows a growth that resembles the predictions of the $\xi - q$ model (Figure 6a). Note, however, that none of these experiments take into account the discharge being contributed from rainfall/runoff in the sedimentary system,

450 i.e., the discharge is set at the left boundary. Differences between the two experimental setups include the dimensionality (1D for the experiments of Guerit et al. (2014) and 2D for those of Rohais et al. (2012) and Whipple et al. (1998)) as well as the nature of the flow (laminar in the Guerit et al. (2014)'s experiments and turbulent in the other two).





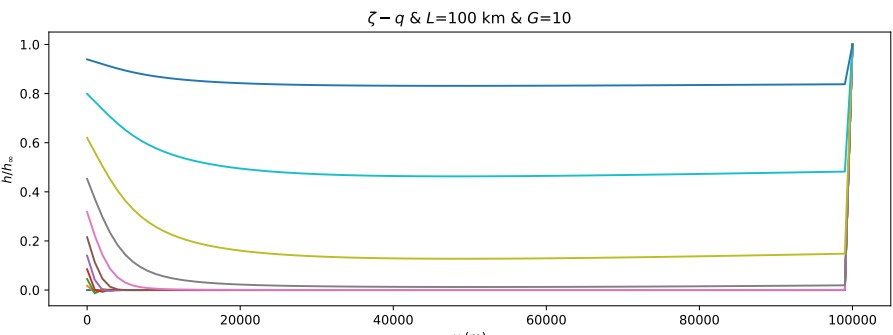

**Figure 17.** Evolution of the sedimentary system for $L > L_0$ using the $\xi - q$ model but a value of $G = 10$. The solution looks very similar to that obtained by using the $TL$ model (Figure 7f).

## 4.6 Value of $G$

All the experiments we have performed with the $\xi - q$ model used a value of $G = 1$. As shown by Guerit et al. (2019), observations from natural sedimentary systems suggest a range between 1 and 2 for $G$. It is also in this range that the $\xi - q$ model shows the most interesting behavior. For values of $G >> 1$, the model tends to behave exactly like the $TL$ model with, for example, an identical dependence of the response time on system length and a geometrical evolution that is identical to that of the $TL$ model as shown in Figure 17. For values of $G << 1$ the $\xi - q$ model predicts that the transfer system is very small, i.e., the volume of sediment that it can store is negligible. This would lead to fan slopes that are much lower than observed in nature.

## 4.7 Hack's law in a depositional system and optimum values of $m$ and $n$

In setting up our experiments, we have assumed, for both models, that Hack's law applies to depositional systems. Edmonds et al. (2011) showed that even low slope, depositional environments such as deltas obey Hack's law with an exponent ($p$) very close to 2. We have also checked that this holds using a 2D landscape evolution model that solve the $\xi - q$ equation based on the algorithm developed by Yuan et al. (2019). The model geometry is of a sediment/water point source feeding material over a flat area of dimension 100x100 km. Three experiments were performed assuming upstream drainage areas, $A_0$, of $10^7$, $10^8$ and $10^9$ m$^2$, respectively. We show in Figure 18 the geometry of the 10 longest channels originating from the center of the model where the sediment/water flux is imposed (panel a), as well as the relationship between distance to the source (center of the model) and drainage area (panel b). We see that, in all three experiments, the most active channel sees its distance-drainage area relationship smoothly transition from $A_0$ to a relationship described by Hack's law with an exponent of 2. Most other channels that form along the sides of the fan and flow unto the edges of the model follow Hack's law with an exponent of 2.

If our interpretation is correct, it implies that to reproduce the observed linear relationship between upstream drainage area and fan size that the value of $p$ must be close to 2 even in the fan where, by definition, the system traverses the transition





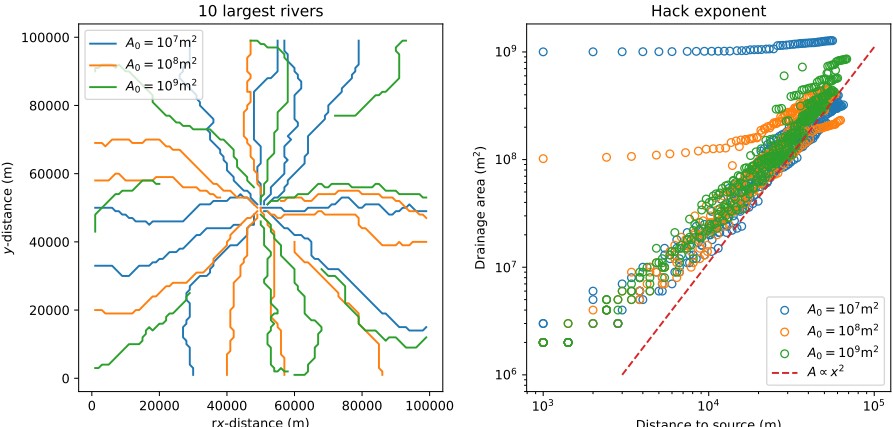

**Figure 18.** Results of three two-dimensional landscape evolution models where sediment and water are provided at a rate proportional to the surface area, $A_0$, of an assumed upstream catchment at the center of the model leading to the formation of a conic sedimentary system. The three models correspond to three different values of $A_0$. a) geometry of the 10 major rivers for each of the three models and b) computed relationship between drainage area and distance to source for those 10 major rivers; the red dashed line has a slope corresponding to an exponent of $p = 2$ in Hack's law.

between confined to unconfined water flow. We have shown (Figure 4) that the partitioning of sediment between the fan and

475    the alluvial plain is determined by the value of the ratio $m + 1/n$. To obtain a significant break in slope between the fan and the alluvial plain, i.e., as is observed in natural systems (Blair and McPherson, 2009), the ratio $m + 1/n$ must be in the range [1 to 2] (see Figure 4). This, in turn, implies that the most likely values of $m$ and $n$ are in the range [1 to 1/3] and [2 to 2/3], respectively, as the concavity of river channels implies that $m/n \approx 0.5$. Of course this is only valid if we wish to have a representation of both the orogenic and depositional parts of the system with a unique set of exponents, an objective that may

480    only be realistic in the context of a reduced-complexity model that is designed to reproduce the long-term and system-scale features of the source-to-sink system, and not the details of the physical processes at play.

### 4.8    Residence time

Both equations used here to model sediment transport are expressed in an Eulerian framework, i.e., using a frame of reference that is fixed with respect to the system's boundaries. Such an approach does not permit to easily track sediment particles and

485    estimating their residence time inside the fan/alluvial plain system as done by Carretier et al. (2016). An alternative approach consists in approximating the residence time, $\tau_R$, by the turnover time that is defined as the ratio between the volume of the active part of the transporting system, $V_a$, and the imposed sediment flux, $q_0$:

$$\tau_R = \frac{V_a}{q_0} \tag{37}$$





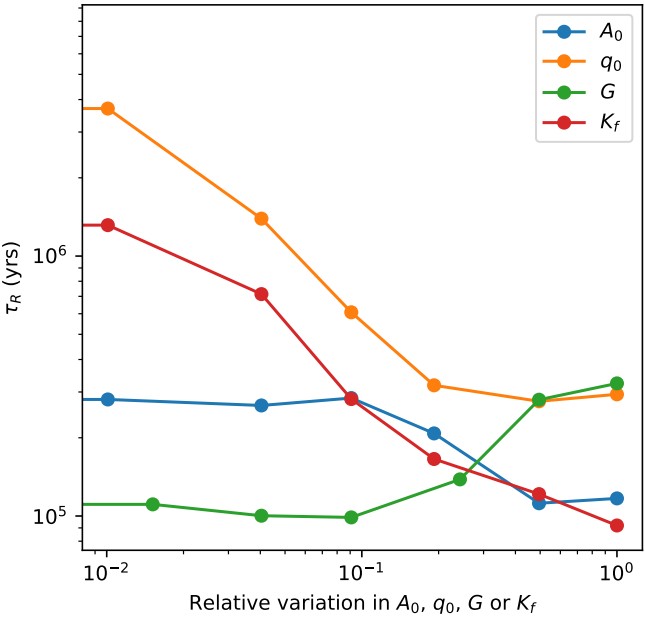

**Figure 19.** Residence time computed using equation 37 for the following range of model parameters: $A_0 \in [10^7 - 10^9]$ m$^2$, $q_0 \in [10^4 - 10^6]$ m$^3$/yr, $G \in [0.02 - 2]$ and $K_f \in [10^{-6} - 10^{-4}]$.

At steady-state $V_a$ is the integral over the sedimentary domain of the thickness of the active layer, $h_a(x,y)$, that can be approximated by the standard deviation of the surface topography over many time steps. This can only be computed using the 2D model where avulsions affect the upper layers of the model. We show in Figure 19 computed values of this residence time as a function of the main model parameters, $A_0$. $q_0$, $G$ and $K_f$ for a 2D model setup similar to the one used in the previous section, i.e., for model parameters and size identical to those used for the model run shown in Figure 18. We see that the residence time varies between $10^5$ and $10^6$ years. More importantly, the model predicts that the residence time varies as $q_0^{-1}$ and $K_f^{-1}$. It also increases linearly with $G$ for values of $G > 10^{-1}$ and is varies with upstream catchment area to the power $-m$, $A_0^{-m}$, as expected.

### 4.9 Effect of basin subsidence on fan size/shape

All results shown so far assume that there is no subsidence in the depositional area. However, most regions adjacent to a mountain belt (or sediment source) experience syn-orogenic subsidence likely driven by flexural isostasy. It has been suggested that the pattern of this subsidence exerts a strong influence on the shape of the resulting alluvial fan (Paola et al., 1992; Parker et al., 1998). We tested the influence of basin subsidence on the shape of the depositional system by running numerical experiments similar to the reference model presented in Figure 3 but adding a subsidence term of the form:

$$s = -s_0(1 - e^{-\alpha x/L}) \tag{38}$$





to both equations 3 and 7. $s_0$ is the maximum subsidence rate at the mountain front and $\alpha$ controls the rate of change of the
subsidence with distance away from the mountain front, $x$. Large values of $\alpha$ correspond to a large rate of change in subsidence
and thus a narrow area of concentrated subsidence near the mountain front, whereas small values of $\alpha$ correspond to a broad
area of subsidence.

In Figure 20a, we show the results of three numerical experiments in which we vary the subsidence rate by 2 orders of
magnitude for a value of $\alpha$ of 7 for an open system (i.e., where $L >> L_0$). In Figure 20b, we show the results of another set
of three experiments in which $\alpha$ is varied between 3 and 10. We see that for all values of the subsidence rate and extent, the
shape of the fan/alluvial plain system is only mildly affected by the imposed subsidence. The sharp transition in slope between
the fan and the alluvial plain at the location $x = L_0$ is preserved. For constrained systems (Figure 20c and d), the shape of the
system is more strongly impacted by the subsidence. The extent of the fan is reduced when the subsidence is fast but the extent
of the subsidence function does not seem to matter much. Interestingly, in all cases, the slope of the fan is not affected by the
subsidence. This demonstrates that in a sedimentary system that sees discharge increase with distance from the mountain front,
the size and extent of the fan, or where it connects to the alluvial plain, are only marginally controlled by the subsidence rate
or extent of the underlying basement. This results applies equally to both the $TL$ and $\xi - q$ models.

## 5 Conclusions

The work we presented, while focused on determining the similarities and differences between the $\xi - q$ and $TL$ models, led
us to present a new analytical solution for the steady-state shape of depositional systems fed by an orogenic system. We have
shown that both models yield the same steady-state solution and that the resulting 1D profile predicts the first-order morphology
of depositional systems and explains key observations made on the size and slope of alluvial fans.

From the two basic evolution equations we have also extracted expressions for the response time of sedimentary systems and
shown that for model parameter values that lead to the same steady-state solution, the two models predict different response
times and, most importantly, different dependencies on system length. The $\xi - q$ model is, in general, characterized by longer
response times than the $TL$ model by a factor that depends on the ratio of the system drainage area to the area in active sediment
transport.

This implies that a proper understanding and parameterization of flood plain width is essential to better quantify the differences between the two models. A potential avenue for this is to use 2D versions of the two models that incorporate a proper
dynamic prediction of flood plain width, which, in turn, requires at minima the use of the shallow water equation. Such models
exist (Simpson and Castelltort, 2006; Davy et al., 2017, for example) but have not been used to perform this scaling analysis
yet.

Using multi-direction flow routing algorithms in landscape evolution models that do not use the shallow water approximation
and therefore imply a simple relationship between flood-plain width and discharge, could be useful as finite width (i.e., larger
than the unit spatial discretization) seems to emerge from these models.

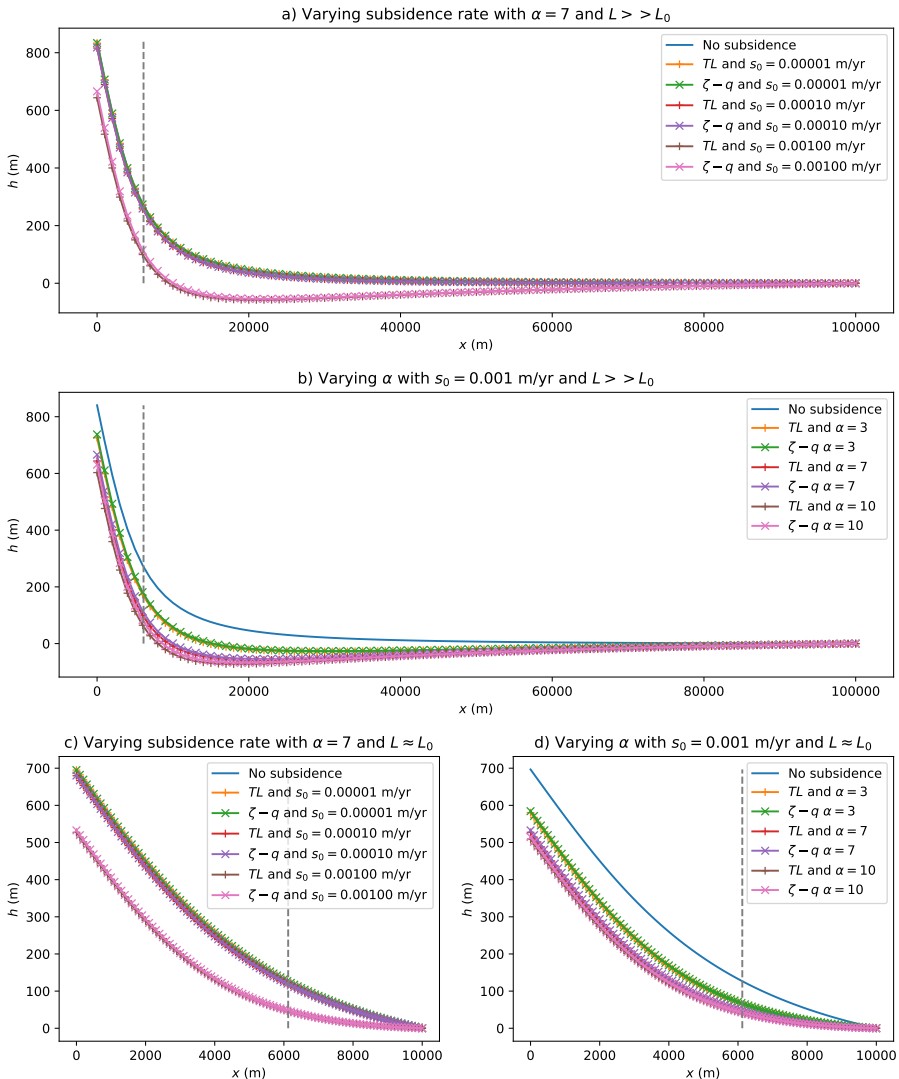

**Figure 20.** Numerical model experiments comparing the steady-state solutions of modified versions of equations 3 ($TL$ model) and 7 ($\xi - q$ model) in which subsidence is imposed at a rate $s_0$ and over an extent controlled by $\alpha$ (see text for details) to the solution without subsidence.

In Figure 21, we show 2D numerical simulations performed by using the FastScape algorithm (Braun and Willett, 2013) combined with Yuan et al. (2019)'s implicit implementation of Davy and Lague (2009)'s $\xi - q$ algorithm and a multi-direction flow algorithm where discharge and erosion rate are distributed according to a set power of local slope (here 1). The model setup is made of a flat plane subject to a constant sediment flux and discharge from a point located in the middle of one of its boundaries. The exponent values are $m = 1$ and $n = 2$. The model broadly reproduces the 1D steady-state solution and, most importantly, its dependence on upstream drainage area. As shown in Figure 22, most of the deposition in the fan area takes


Earth **Surface**
Dynamics
Discussions

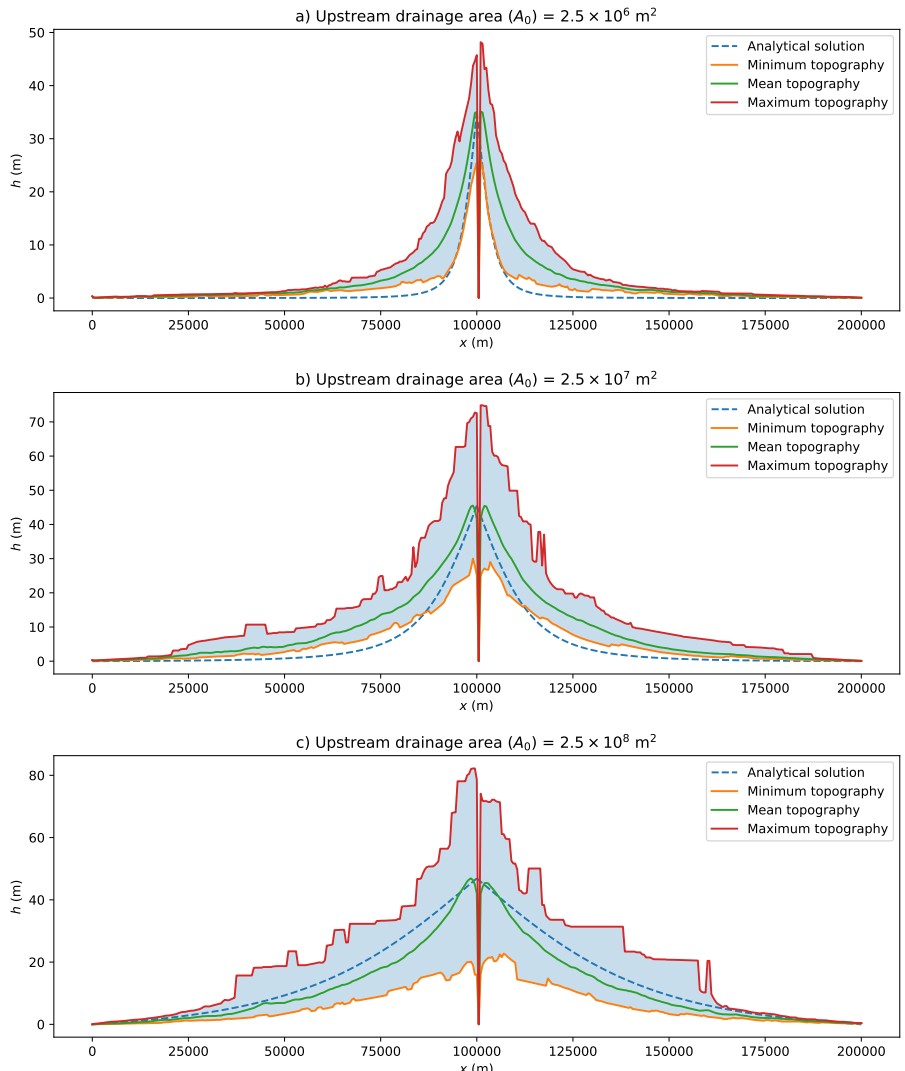

**Figure 21.** Comparison between the 1D analytical solution (Equation 16) and the results of a 2D model for three values of the upstream drainage area $A_0$.

place along the main channel, i.e., the channel that receives both the sedimentary flux and the contribution to its drainage area from the hypothetical upstream catchment. In other parts of the system, deposition is focused along secondary channels while erosion is more uniformly distributed.

Much work is however necessary to better characterize the transient behavior of this model, and in particular, how channel (or flood-plain) width is set (it is definitely greater than the unit spatial discretization but does not scale linearly with spatial resolution) and what determines the frequency of avulsions.





**Appendix A: Numerical method to solve the two equations assuming $n = 1$**

For the *TL* equation, we used a second-order accurate centered scheme to approximate the spatial derivatives and a first-order

accurate implicit scheme to approximate the time derivative. For the $\xi - q$ equation, we used a first-order accurate scheme to approximate the spatial derivative, a first-order accurate implicit scheme to approximate the time derivative, and the rectangle rule to estimate the integral. This yields the following discretized forms:

$$\frac{K_d \nu^{m+1} \Delta t}{\Delta x^2} \left[ A_{i-}^{m+1} h_{i-1} + (1 + A_{i+}^{m+1} + A_{i-}^{m+1}) h_i + A_{i+}^{m+1} h_{i+1} \right] = h_{i,0} \text{ for } i = 2, \cdots, n_x - 1 \tag{A1}$$

for the *TL* equation, where $Ai+ = (A_i + A_{i+1})/2$ and $A_{i-} = (A_i + A_{i-1})/2$, $h_i$ is current topographic elevation at node $i$ and

$h_{i,0}$ is the topographic elevation at the same node at the previous time step, $\Delta x$ is the distance between two nodes, $\Delta t$ is the time between two time steps and $n_x$ the number of nodes used to discretized the river, and:

$$(1 + \frac{K_f \nu^m \Delta t}{\Delta x} A_i^m) h_i - \frac{K_f \nu^m \Delta t}{\Delta x} A_i^m h_{i+1} + \frac{1}{\xi A_i \nu} \sum_{j=1}^{i-1} h_j = h_{i,0} + \sum_{j=1}^{i-1} h_{j,0} + \frac{\Delta t}{\xi A_i \nu} q_0 \text{ for } i = 1, \cdots, n_x - 1 \tag{A2}$$

for the $\xi - q$ equation, where $q_0$ is the incoming sediment flux (expressed in m yr$^{-1}$).

These systems of equations can be written in matrix form:

$$A_d H = B_d \text{ and } A_a H = B_a \tag{A3}$$

where $A_d$ and $A_a$ are square matrix of dimension $n_x \times n_x$ and $B_d$ and $B_a$ are vectors of dimension $n_x$. $H$ is the solution vector containing the topographic elevation of the nodes. For simplicity, we use a simple general direct solver for these two systems of algebraic equations even though $A_d$ is a tridiagonal matrix and $A_a$ is a Hessian matrix.

**Appendix B: Response time scaling for various values of $m$ and $n$**

In Table **??**, we illustrate this scaling for commonly assumed values of $p$, $m$ and $n$. We consider a linear case ($n = 1$) and a non-linear case ($n = 2$).

**Appendix C: Validation of response time scale relationship**

In the first set of experiments, we vary the erodibility, $K_f$, in the $\xi - q$ equation and the transport coefficient, $K_d$, in the *TL* equation. The results are shown in Figure C1 and demonstrate that both response times varies as the inverse of the diffusivity

or erodibility, as predicted by Equations 23 and 27.

In a second set of experiments, we varied $G$, which yielded the expected scaling in the $\xi - q$ model as shown in Figure C2.






**Table B1.** Scaling of the *TL* and $\xi - q$ response times, $\tau_{TL}$ and $\tau_{\xi-q}$, with the various forcings and parameters for two sets of values of $m$ and $n$. We consider a linear case ($n = 1$) and a non-linear case ($n = 2$) but keep the ratio between $m$ and $n$ at 0.5. For both cases, we use $p = 2$.

| | | | $L$ | $q_0$ | $K_d$ or $K_f$ | $w$ | $A_0$ | $G$ |
|---|---|---|---|---|---|---|---|---|
| $\tau_{TL}$ | $m = 0.5$ | $L \leq L_0$ | 2 | 0 | $-1$ | 1.5 | $-1.5$ | - |
| | & $n = 1$ | $L > L_0$ | $-1$ | 0 | $-1$ | 1.5 | 0 | - |
| $\tau_{TL}$ | $m = 1$ | $L \leq L_0$ | 2 | $-0.5$ | $-0.5$ | 1 | $-1$ | - |
| | & $n = 2$ | $L > L_0$ | $-2$ | $-0.5$ | $-0.5$ | 1 | 1 | - |
| $\tau_{\xi-q}$ | $m = 0.5$ | $L \leq L_0$ | 1 | 0 | $-1$ | 0.5 | $-0.5$ | 0 |
| | & $n = 1$ | $L > L_0$ | 0 | 0 | $-1$ | 0.5 | 0 | 0 |
| $\tau_{\xi-q}$ | $m = 1$ | $L \leq L_0$ | 1 | $-0.5$ | $-0.5$ | 0 | 0 | $-0.5$ |
| | & $n = 2$ | $L > L_0$ | $-1$ | $-0.5$ | $-0.5$ | 0 | $-1$ | $-0.5$ |





**Figure 22.** Results from a 2D numerical experiment corresponding to the solution shown in Figure 21a. a) Surface topography, b) logarithm of drainage area and c) deposition/erosion rate for the last time step of the model. Light colors/positive values in panel c correspond to deposition. The black line is the zero contour.





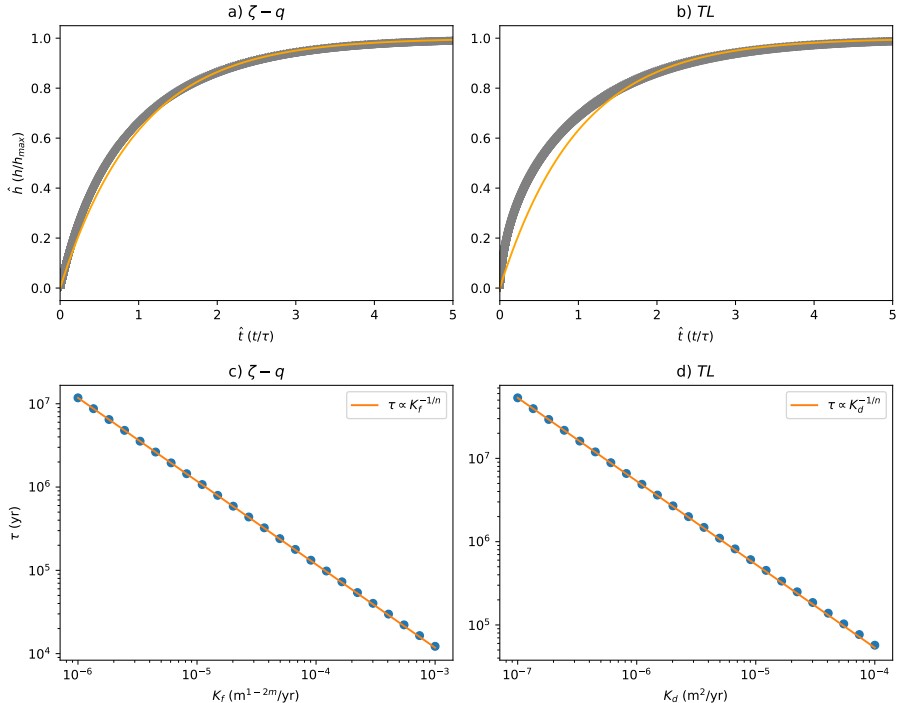

**Figure C1.** Computed response times for 24 numerical experiments in which the erodibility, $K_f$, or the diffusivity, $K_d$ ,of the model were varied.

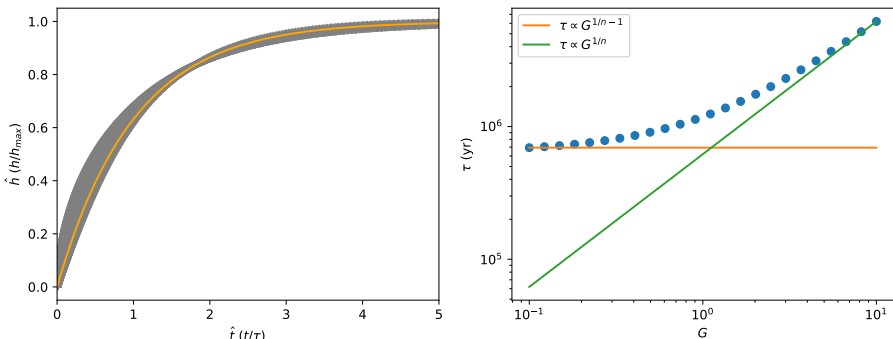

**Figure C2.** Computed response times for 24 numerical experiments in which the deposition constant, $G$, of the $\xi - q$ model was varied.



## Appendix D: Expressions for the flux

For the *TL* equation:

$$q_{TL} = \frac{K_d}{w}(A\nu)^{m+1}|S|^n \tag{D1}$$

and for the *TL* equation:

$$q_{\xi-q} = \frac{K_f}{Gw}(A\nu)^{m+1}|S|^n + \frac{A\nu}{Gw}\frac{\partial h}{\partial t} \tag{D2}$$

## Appendix E: Geometrical interpretation of the response time

The time to fill a triangle of height $h_0$ and length $L$ with an incoming sedimentary flux $q_0$ is:

$$\tau_{fill} = \frac{h_0 L}{2q_0} = q^{1/n-1}K_d^{-1/n}w^{(m+1)/n}A_0^{-(m+1)/n}L^2/2 = \tau_{TL}/2 \tag{E1}$$



*Author contributions.*  J Braun performed all the work described in this manuscript and prepared it.

     *Competing interests.*  None

     *Acknowledgements.*  The author wish to thank S. Castelltort and S. Carretier for comments made on earlier versions of this manuscript.



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
