# Peer review of "Comparing the transport-limited and $\xi-q$ models for sediment transport"

_Earth Surface Dynamics, 2021_

## Referee Comment (RC2)

[referee-annotated manuscript omitted]

---

## Author Response (AR1)

**Response to reviewers comments and suggestions**

**Jean Braun**

First of all, I wish to thank the two reviewers and Stefan for their thorough reviews and comments. They have certainly helped me improve the manuscript. I especially thank Alex Densmore for suggesting how to improve its focus. I have followed his advise and remove the part of the discussion dealing with 2D models which will be part of a more detailed manuscript to be submitted at a later stage.

In preparing the revised version I believe I have addressed all of their suggestions/comments/concerns. I now detail my responses and detail the modifications made to the manuscript which are minor but pervasive. I have also attached a version of the manuscript where the changes have been highlighted.

**1 Sebastien Carretier's review**

This manuscript addresses the general problem of the propagation of a sedimentary signal between a source and a basin. Jean Braun compares the predictions of two theoretical sedimentary basin topography models, one (old) corresponding to the transport-limited model and the (more recent) E-q model of Davy and Lague (2009) which belongs to the family of under-capacity models. It shows clear differences of the E-q model compared to the TL model, among which the possibility of transmitting all the frequencies of variations from the source to the basin. It is also clear that the response time is longer with the E-q model than with the TL model and its different scaling with river length is also non intuitive. Finally, this analysis shows that both models result in the same equilibrium topographic profile, with a linear upstream portion and a more distal concave portion, which results from the downstream increase of runoff by accumulation of rainfall on the piedmont. The manuscript is dense but very clear and well illustrated. It will be of interest to a broad community of sedimentologists and geomorphologists working on source-to-sink issues, shedding new light on the dependence of the interpretation of the sediment record on the transport model in the transfer zone. I have already read and commented on a pre-submission version of this manuscript. I feel that the present version is almost in a good shape, although I have several questions:

**Response**

*Dear Sebastien, Many thanks for the very constructive comments and the thorough review.*

**1.1 Point 1**

The demonstration that the shape of the topographic profile of the foothills has two parts, one linear and the other concave, is an advance that allows a better understanding, for example, of the observed scaling relationship between alluvial fan surface and watershed surface. However, river slope also depends on grain size in nature and in theory (e.g. Parker et al., 1998). Preferential deposition of coarse sediment upstream can influence the variation of this slope. The experiments of P. Delorme (Delorme et al., Earth Surf. Dynam., 5, 239-252, 2017) using two grain types of different mobilities show a longitudinal slope transition associated with the deposition of these two materials. Since grain size decrease is observed in natural systems, I wonder what the influence of downstream grain size variation on Lo might be in the model.

**Response**

*As indicated by Sebastien, it has been shown that grain size exerts a strong control on sedimentary system slopes and is called upon as the mechanism/process that controls the break in slope observed at (or that defines) the foot of sedimentary fans. It therefore is interesting to note that a model that does not incorporate a transport capacity/efficiency dependence on grain size is able to produce a break in slope that agrees with the observed scaling in natural systems. So, rather than hypothesizing on what would be the effect of introducing a grain-size dependence, I added a short paragraph to the discussion section making that point.*

**Change in the manuscript**

I added the following paragraph in the *New analytical solution* section:

*Experimental work suggests that the break in slope at the foot of a sedimentary fan is a result of grain size control on transport efficiency (Parker et al., 1998). Interestingly, I show here that the break in slope can be produced with a model that does not include a grain-size control on transport coefficient ($K_d$) or depositional parameter ($G$). Because the model produces the observed area and slope scalings with upstream catchment area (something that cannot be derived from the grain size dependence on transport efficiency alone), I would like to suggest that the observed transition in grain size at the foot of sedimentary fans may be a consequence of the change in transport efficiency rather than the cause of it. But this remains to be tested, potentially by performing laboratory experiments that consider rainfall accumulation and contribution to discharge within the sedimentary system.*

**1.2  Point 2**

I am probably influenced by arid environments, but in these settings, rainfall and evaporation distribution may be strongly influenced by elevation, so that the runoff does not increase significantly along the piedmont and further downstream. This would correspond to a nearly constant drained area Ao in the model. This would influence the shape of the equilibrium profile, and perhaps the transient dynamics. Is it possible to discuss this from the scaling relationships established in this theory?

**Response**

*The shape of the steady-state solution is strongly dependent on how rainfall accumulates in the depositional area. Here we have assumed that the rainfall is uniform and, to arrive at the scaling that the fan area scales linearly with the upstream drainage area requires that rainfall is similar in both the orogenic and depositional areas. One easy way to test this hypothesis is to see how the steady-state solution depends on Hack's k constant. I have added a figure in which I display how the shape of the system depends on k and p. For a reduced rainfall rate in the basin (low k values), the fan becomes wider whereas for an increased rainfall rate in the basin, the fan becomes narrower.*

**Change in the manuscript**

I added the following paragraph in the *Hack's law in a depositional system and optimum values of m and n* section:

*To further illustrate this last point, I computed the effect of varying both Hack Law's parameters (k and p) on the shape of the steady-state solution. The results are shown in Figure 19 and show that varying the rainfall rate (or changing the value of k) in the basin area (compared to the orogenic area) results in a wider fan for greater values of k, and vice-versa. Changing p also affects the fan steepness. Lower p values (compared to 2) leads to a much reduced slope contrast between the fan and alluvial plain areas.*

**1.3 Point 3**

The weak influence of subsidence on the longitudinal profile at equilibrium is clearly demonstrated. However, during an increase in sediment flux or a change in precipitation, part of the flux would be trapped by subsidence. I anticipate that this will not fundamentally change the rather "upward" or detachment-limited behavior of the E-q model compared to the "downward" behavior of the TL model, but can it change the conclusions regarding the transmission of all frequencies of source variations in the E-q case?

**Response**

*Sebastien raises an interesting point. I have made a few tests and, in terms of signal transmission, the two models appear to behave similarly with or without subsidence. I feel, however, that a more complete study on how subsidence affects the transmission of signals through the system is necessary to properly answer his question. As the manuscript is already very long and already contains much material to be digested by the reader, I decided it would be better to present these results separately, if they end up being important.*

**1.4 Point 4**

A constant width w is assumed in the model. Is there a way to anticipate how possible variations of w during sediment flux or rainfall variations (with potential entrenchments and infill) may affect the transfer of source signals?

**Response**

*I have already indicated how a spatially variable width would affect the model equations at the end of the section describing the two models. From this, there is no indication that the model response would be different. However, I have added an additional reference to recently published work by Goldberg et al, 2021 in JGR where this assumption is introduced and justified.*

**Change in the manuscript**

I added a reference to Goldberg et al (2021) at the end of the models description section.

**1.5 Point 5**

Given the density of information in this manuscript, I wonder if a summary table or diagram showing the differences between the two models would not be useful to facilitate dissemination to a large audience.

**Response**

*I have added such a table*

**Change in the manuscript**

I added the following table to the conclusion section.

**1.6 Point 6**

The Fastscape simulations are very promising. In my own experience of such kind of simulations with CIDRE using also the E-q model and multiple flow algorithm, the rivers move a lot on the piedmont, as

Table 1: Comparison between the two models.

| Model | $\xi - q$ | $TL$ |
|---|---|---|
| Steady-state solution | Slope change between fan and alluvial plains | Identical |
| Growth style | From toe to apex | From apex to toe |
| Flux evolution | Instantly finite and $= q_0$ at apex | Grows from 0 to $q_0$ |
| Response time, $\tau$ | Longer | Shorter |
| $\tau$ dependency on model parameters | $L^1$ for $L \leq L_0$ | $L^2$ for $L \leq L_0$ |
| | $L^{1-mp}$ for $L > L_0$ | $L^{2-p(m+1)}$ for $L > L_0$ |
| | $q_0^{1/n-1}$ | Identical |
| | $K_f^{-1/n}$ | $-$ |
| | $G^{1/n-1}$ | $-$ |
| | $-$ | $K_d^{-1/n}$ |
| | $A_0^{1-(m+1)/n}$ for $L \leq L_0$ | $A_0^{-(m+1)/n}$ for $L > L_0$ |
| | $A_0^{-(m+1)(n-1)/n}$ for $L \leq L_0$ | $A_0^{(m+1)(n-1)/n}$ for $L > L_0$ |
| Periodic variations in input flux | Signals with periods shorter than response time are dampened but transmitted | Signals with periods shorter than response time are NOT transmitted |
| | Flux signals are transmitted without major changes in topography | Flux signals are transmitted through local topographic changes |
| Periodic variations in precipitation rate | Signals with periods longer than response time are not transmitted | Identical |

observed in Fastscape, with avulsions and ephemeral confluences in particular when adjacent catchments in the mountain feed adjacent fans in the basin that merge downstream, which is different from a one-point source. I wonder if this dynamics can be merged along a line in average, but as Jean Braun writes in the conclusion, this is for another story.

**Response**

*I agree with Sebastien that this is another story. A thorough investigation is needed and cannot be included in this already quite long manuscript. Following Alex Densmore's suggestion (see below), I have in fact removed some of the 2D material.*

**1.7  Specific comments**

1. Line 51 : latex tipo Theta
   *Corrected*

2. Line 197 parenthesis lacking
   *Corrected*

3. Line 220 is it and = or a propto in this equation?
   *Corrected*

4. Line 240 I do not understand where does this formula of L come from.
   *To obtain this, one needs to find the minimum of the function in equation 26 by finding the root of its derivative with respect to L; this is a bit tedious to put in the text of the manuscript but is standard procedure.*

5. Lines 241-245 Could you define a bit more what you mean by linear and non-linear systems here?
   *I meant regardless of the value of $n$. Corrected.*

6. Line 445 about the Whipples experiments :It seems to me that it corresponds better to the TL model in Figure 6d rather than the E-q model in Figure 6a... Maybe the experiments of P. Delorme et al. Growth and shape of a laboratory alluvial fan, Physical Review (2018) may help the discussion here.
   *I disagree: the important feature of the Whipple et al experiment that resembles the $\xi - q$ model is how it responds to variations in flux. I changed the text accordingly.*

7. Line 485 : I think the correct paper here is Carretier et al. Earth and Planetary Science Letters 546, 116448 (2020).
   *Corrected*

8. Line 565 tipo in Tables reference
   *Fixed*

9. Figure 20 caption: indicate what the dashed line is
   *Corrected*

10. In many figures there is tipo on the xi-q model (zeta-q is used instead)
    *Fixed*

**2  Stefan Hergarten's question**

Dear Jean,

A very interesting paper – I am actually working on similar stuff. I recently wrote about knickpoints (doi 10.1029/2021JF006218), where I had some discussion with a reviewer about sediment deposition. Can we

"extrapolate" the undercapacity model (or the equivalent formulation as shared stream-power model which I use) "symmetrically" (so with the same parameters) to the regime of deposition? Wouldn't there be an alluvial cover with an erodibility Kf much higher than that of the bedrock? If so, the model would come very close to the transport-limited model in the depositional regime. I remember that the SPACE model by Charles Shobe et al. (doi 10.5194/gmd-10-4577-2017) implements something in this direction. If I was a reviewer, I would ask for some more discussion of this aspect. Anyway, I would be very happy to get your opinion on this aspect.

Best regards, Stefan

**Response**

*Dear Stefan, Many thanks for the comment/question.*

*In this manuscript, I wish to remain in a purely sedimentary setting where the 'basement' is itself made of previously deposited sediment. In such a setting it is not necessary to take into account spatial/temporal variations in erodibility. However, I address the issue of how the under-capacity model (and especially its response time scale) varies as a function of the erodibility. It would definitely be interesting to see how a 'mixed' system (where bedrock is intermittently exposed) would behave; i.e., how would the steady-state solution and the response time scale scaling be affected. In particular, one could envisage situations/settings where the relative importance of the erosion/deposition terms differs for bedrock vs. sediment, such that the scaling on $G$ ($1/n$ or $1/n-1$; see Figure C2 in the appendix) is not the same for bedrock and sediment patches.*

**3 Alexander Densmore's review**

This is a well-written and elegant manuscript comparing two common numerical approaches to the problem of how to model the evolution of a sediment routing system. While the approaches themselves are not new, the framework in which the author compares them is, and that framework allows him to demonstrate both fundamental similarities and important differences between the two approaches. There are some excellent insights and predictions that emerge from the work, and I have no doubt that this manuscript will stimulate considerable interest and further work, both on further refinements to these models and on more detailed comparisons with natural systems.

The manuscript is quite concise, bordering on sparse in places, and there are a few elements that would benefit from a bit more explanation or clarity. I've returned an annotated PDF with some queries and suggested edits, all of which are fairly minor. I won't repeat them here except to highlight two points in particular.

**Response**

*Dear Alex, Many thanks for your supportive comments and the thorough review.*

**3.1 Point 1**

First, I think the introduction would benefit from a more explicit statement about the part of a sediment routing system that is under investigation here. The motivation on line 42 is to understand model behaviour 'in a purely depositional environment', but actually the author is applying the model to a very specific setting and some of the assumptions depend on that setting. Details of the application emerge over the following pages, but to avoid confusion I think it would be good to bring this up front; compare, for example, the specificity in the first line of the discussion with the almost offhand phrasing of lines 54-55.

**Response**

*I have modified the last paragraph of the introduction to be more specific on the type of sedimentary systems that the manuscript and its findings are relevant to.*

**Change in the manuscript**

The last paragraph of the introduction now reads:

*Although the purpose of this work is to compare the general behavior of two sediment transport models, I will focus on sedimentary systems that develop at the foot of an orogenic are, more precisely the fan and neighboring alluvial plain. The idea is to study a system that is familiar to sedimentologists but relatively simple in its setting, such that the intrinsic behaviors of the two models can be efficiently analyzed and compared to observational constraints.*

**3.2 Point 2**

Second, there is some new material that comes into the conclusions, including a 2D implementation of one of the models in FastScape and two different fan-building experiments (Figs 21 and 22). None of these are well-explained or justified, and they come as a surprise to the reader. I'm not sure what additional information is gained by the inclusion of this new material, or how it links to the fundamental comparison that is the heart of the manuscript. Thus, my suggestion would be either to integrate this material more clearly into the overall framework (which means introducing it as part of the opening sections) or removing it altogether.

**Response**

*I agree with Alex's suggestion that the presentation of this material needs much greater care. I have therefore removed it and started to prepare another manuscript on the topic.*

**Change in the manuscript**

The paragraph and the two figures have been removed.

**3.3 Minor corrections suggested in the edited version of the manuscript returned by Alexander Densmore**

1. First person plural has been replaced by first person singular all through the manuscript

2. Abstract has been amended as suggested

3. Line 53: Other parts of the introduction have been modified in response to Point 1.

4. Line 104: The question concerning the model setup, the applicability of Hack's law in the sedimentary system and the potential effect of the aridity of the depositional part of an orogenic system have been addressed in response to Sebastien Carretier's similar remarks.

5. Figures 6, 7 and others: Figures have been amended so that the color scheme corresponding to time evolution is made progressive rather than random.

6. Line 200: I have updated the paragraph explaining the difference in response between the two models following a step perturbation in sediment flux to make it clearer.

7. Line 205: It is true that if one changes a parameter of the system to make it transition between two steady-states, the response time might be different if measured/defined in the initial or final stage but this does seem like a minor point compared to finding out what the response time is for a system defined by a defined set of parameters, as is the case when assumed that the system evolves from a nil initial condition.

8. Figure 10 caption has been made clearer.

9. Line 246: I have removed the statement relative to precipitation rate as it should only refer to variations in precipitation rate (not the absolute value) and, as shown by Alex, is therefore misleading.

10. Throughout the manuscript I have made sure that the term 'phase shift' is used rather than 'phase' or 'time lag'.

11. Line 372: the alluvial fan problem has been better described, although in a very succinct manner.

12. Line 410: changes made following Alex's suggestion.

13. Line 419 has been changed

14. Line 432: sentence has been made more general

15. Line 443: I find it difficult to cite all papers reporting experimental work on fan development. Key is that I could not find any where the discharge of rivers in the sedimentary system was contributed to by distributed rainfall. So I made the choice of only displaying a few (and only quoting them) that showed a clear picture of the internal evolution of the system, to help for another point in the discussion.

16. Figure 17 (as well as 7 and 9): corrected

17. Line 470: the point made by Alex has been addressed in my response to Sebastien's question on the same issue.

18. Figure 20: caption modified.

19. Line 495: precision added.

20. Comments on the effect of subsidence in the basin: as explained in my response to Sebastien, I find it difficult to cover all possible patterns of subsidence; the important point is that the steady-state solution of a steep fan adjacent to a lower slope alluvial plain is maintained and that the location of the transition between the two is not greatly affected by the shape of the imposed subsidence curve.

21. Line 527: I have renamed the section 'Conclusions and perspectives' as I believe that the material that follows the conclusions *per se* is fit to be included as perspectives or extensions of the detailed analysis presented here into 2D. I have, however, followed the advise from Alex and removed the figure 21 and the paragraph describing it. I intend to present the results it contains in a more thorough manner in a future manuscript.

22. In Figure C1/C2 and 10, the orange curve represent the exponential curve to which all other (grey) curves have been compared to determine the response time scale. Because it is presented using dimensionless variables along both axis, there is only one such curve. I hope this make more sense. I have tried to better explain this in figure 10 caption.

23. Appendices D and E: I feel that there are already so many mathematical expressions given in the body of the manuscript that even adding a few would render it difficult to 'digest' for the readers.